# MAVS recruits multiple ubiquitin E3 ligases to activate antiviral signaling cascades

Siqi Liu[1][†], Jueqi Chen[1][†], Xin Cai[1], Jiaxi Wu[1], Xiang Chen[1,2], You-Tong Wu[1], Lijun Sun[1,2], Zhijian J Chen[1,2]*

[1]Department of Molecular Biology, University of Texas Southwestern Medical Center, Dallas, United States; [2]Howard Hughes Medical Institute, University of Texas Southwestern Medical Center, Dallas, United States

**Abstract** RNA virus infections are detected by the RIG-I family of receptors, which induce type-I interferons through the mitochondrial protein MAVS. MAVS forms large prion-like polymers that activate the cytosolic kinases IKK and TBK1, which in turn activate NF-κB and IRF3, respectively, to induce interferons. Here we show that MAVS polymers recruit several TRAF proteins, including TRAF2, TRAF5, and TRAF6, through distinct TRAF-binding motifs. Mutations of these motifs that disrupted MAVS binding to TRAFs abrogated its ability to activate IRF3. IRF3 activation was also abolished in cells lacking TRAF2, 5, and 6. These TRAF proteins promoted ubiquitination reactions that recruited NEMO to the MAVS signaling complex, leading to the activation of IKK and TBK1. These results delineate the mechanism of MAVS signaling and reveal that TRAF2, 5, and 6, which are normally associated with NF-κB activation, also play a crucial role in IRF3 activation in antiviral immune responses.

## Introduction

The innate immune system deploys germline-encoded pattern recognition receptors (PRRs) to detect pathogen invasion. Pathogen-associated molecular patterns (PAMPs) such as bacterial LPS and viral dsRNA are recognized by PRRs such as membrane-bound Toll-like receptors (TLRs) and cytosolic retinoic-acid-inducible gene-I (RIG-I)-like receptors (RLRs) (*Akira et al., 2006*). Upon activation, the receptors trigger the production of type-I interferons (e.g., IFNα and IFNβ) and other cytokines (e.g., TNFα and IL-6) to rapidly restrict the infection and further activate the adaptive immune system (*Pichlmair and Reis e Sousa, 2007*; *Stetson and Medzhitov, 2006*).

RLRs include RIG-I, MDA5, and LGP2 (*Yoneyama et al., 2004*; *Yoneyama and Fujita, 2008*). All RLRs contain a DEAD/H-box RNA helicase domain, which binds to double-stranded RNA. RIG-I and MDA5, but not LGP2, harbor N-terminal tandem CARD domains that are crucial for triggering type-I IFN production. RIG-I also contains a C-terminal regulatory domain that specifically binds to RNA bearing 5′ triphosphate (*Hornung et al., 2006*; *Pichlmair et al., 2006*; *Cui et al., 2008*). Upon binding to different viral RNA ligands, RIG-I and MDA5 activate another CARD-containing protein, mitochondrial antiviral signaling protein (MAVS, also known as IPS-1, VISA, and CARDIF), presumably through CARD–CARD interaction. MAVS in turn activates the transcription factors IRF3 and NF-κB, leading to interferon induction (*Kawai et al., 2005*; *Meylan et al., 2005*; *Seth et al., 2005*; *Xu et al., 2005*).

Recent studies have provided insights into the mechanisms by which RIG-I and MAVS are activated upon viral infection (*Gack et al., 2007*; *Zeng et al., 2010*; *Kowalinski et al., 2011*; *Luo et al., 2011*; *Jiang et al., 2012*). The sequential binding of RIG-I to viral RNAs and unanchored lysine-63 (K63) polyubiquitin chains promotes RIG-I to form higher order oligomers, which then rapidly induce MAVS polymerization. The MAVS polymers recruit other MAVS molecules on the mitochondrial surface to form larger polymers through a prion-like mechanism (*Hou et al., 2011*). These MAVS polymers potently activate the cytosolic kinases IKK and TBK1 through mechanisms that remain to be elucidated.

*For correspondence: zhijian.
chen@utsouthwestern.edu

[†]These authors contributed
equally to this work

Reviewing editor: Feng Shao,
National Institute of Biological
Sciences, China

**eLife digest** The innate immune system can detect and destroy viruses, bacteria and other pathogens that enter the human body. In particular, inside cells, viral RNA can bind to and activate a protein called RIG-I. This protein switches on another protein, called MAVS, which can activate other copies of itself. These MAVS molecules then aggregate together on the membrane of mitochondria and send a signal that leads to the production of small proteins, called cytokines, which stimulate an inflammatory response and ultimately neutralize the virus.

Although many of the proteins that are activated by MAVS in the innate immunity signaling pathway have been identified, precisely how MAVS transmits this signal is unknown. Now, Liu et al. explore how this protein can propagate signals in the innate immune response by monitoring activation of the transcription factors IRF3 and NF-κB, which transcribe cytokine genes.

Previous studies have suggested that a protein known as ubiquitin is needed to activate RIG-I, and that this protein collaborates with MAVS to signal through the innate immunity pathway. Liu et al. found that a group of proteins including TRAF2, TRAF5, TRAF6 and LUBAC relay the antiviral signal by binding to MAVS. These so-called 'E3 ligases' string ubiquitin together in chains called polyubiquitin, which is essential for activating signaling after, or downstream of, MAVS; however, the association of these E3 ligases with MAVS also requires that multiple copies of MAVS cluster together.

MAVS, the TRAF proteins and LUBAC collectively recruit other innate immunity pathway proteins to activate IRF3 and NF-κB, and thus transcription of the genes that control the innate immunity response. Together, these results show the intricate interplay of proteins needed to eliminate viruses from the body.

The ubiquitin system has been shown to be important for RIG-I signaling at steps both upstream and downstream of MAVS. Upstream of MAVS, polyubiquitin binding to RIG-I is required for RIG-I activation as described above. Downstream of MAVS, K63-linked polyubiquitin and the E2 UBC5 are required for IRF3 activation (*Zeng et al., 2009*). NEMO, the regulatory subunit of IKK and TBK1, functions as a ubiquitin sensor in different pathways that lead to NF-κB and IRF3 activation (*Ea et al., 2006*; *Wu et al., 2006*; *Zhao et al., 2007*; *Zeng et al., 2009*). Additionally, several proteins, including TANK, SINTBAD, NAP1, and STING (also known as MITA, ERIS, and MPYS), have been found to associate with TBK1 (*Sasai et al., 2006*; *Guo and Cheng, 2007*; *Ryzhakov and Randow, 2007*; *Ishikawa and Barber, 2008*; *Jin et al., 2008*; *Zhong et al., 2008*; *Sun et al., 2009*). Several E3 ligases including TRAF3, TRAF5, cIAP1/2, and MIB1/2 were proposed to regulate IRF3 activation downstream of MAVS (*Saha et al., 2006*; *Mao et al., 2010*; *Tang and Wang, 2010*; *Li et al., 2011*; *Wang et al., 2012*). In addition, LUBAC, an E3 ligase complex that synthesizes linear ubiquitin chains, has been shown to negatively regulate the RIG-I pathway (*Inn et al., 2011*; *Belgnaoui et al., 2012*). However, the biochemical mechanism of how ubiquitination regulates the activation of the protein kinases and transcription factors remains unclear.

We have previously described a cell-free system that mimics cellular response to viral infection. To further dissect the mechanism of how MAVS propagates downstream signaling, we performed conventional purification and identified TRAF6 as an IRF3 activator in the presence of activated MAVS. Moreover, by knocking down TRAF6 in *Traf*2/5-deficient cells, we found that TRAF2 and TRAF5 act redundantly with TRAF6 to activate both IRF3 and NF-κB in response to virus. In addition, we provide evidence that the E3 ligase activity of TRAF6 is essential in the TRAF6-dependent activation of IRF3 and NF-κB, whereas the E3 ligase activity of TRAF2 is redundant with that of LUBAC in the *Traf*2-dependent pathway downstream of MAVS. Furthermore, mutations of the binding sites for TRAF2, TRAF5, and TRAF6 on MAVS abolished the ability of MAVS to activate downstream signaling after virus infection, without affecting its ability to form prion-like polymers. MAVS polymerization mutants, however, failed to recruit TRAFs. Finally, we found that through its ubiquitin-binding domains, NEMO formed a ubiquitination-dependent complex with TRAFs and MAVS, both in vitro and in cells. These results demonstrate a key role of TRAF2, TRAF5, and TRAF6, which form a ubiquitin-dependent signaling complex with NEMO and the kinases, in propagating antiviral signaling downstream of MAVS.

## Results

### Identification of TRAF6 as an IRF3 activator

We have previously established a cell-free IRF3 dimerization assay that mimics IRF3 activation in virus-infected cells (*Zeng et al., 2009*). When crude mitochondrial fraction (P5) from Sendai virus-infected HEK293T cells was incubated with cytosolic extracts (S5 or S100) from uninfected cells, along with [$^{35}$S]-IRF3 and ATP, IRF3 dimerization was detected by native gel electrophoresis followed by autoradiography (*Figure 1A,B*). Similarly, IκBα phosphorylation was detected by immunoblotting with a phospho-IκBα specific antibody (*Figure 1B*). P5 from virus-infected cells could also be replaced by purified MAVS without its transmembrane domain (MAVSΔTM) as previously described (*Hou et al., 2011*). MAVSΔTM expressed and purified from *E. coli* forms functional polymers, thereby bypassing the requirement for its mitochondrial membrane localization (*Hou et al., 2011*). In our previous study, HeLa S100 was separated by Q-Sepharose column into Q-A containing the flow-through and Q-B containing proteins eluted with 0.3 M NaCl (*Zeng et al., 2009*). Both Q-A and Q-B were required to support IRF3 activation in our in vitro assay. The key factor in Q-A was identified as the ubiquitin E2 Ubc5 (*Zeng et al., 2009*).

Q-B contains multiple factors known to be important for virus-induced IRF3 activation, such as NEMO, TBK1, ubiquitin, and E1 (data not shown). Q-B also contains the IKK complex; however, IKKα/IKKβ double-deficient MEF cells activated IRF3 normally in response to infection by vesicular stomatitis virus (VSV), an RNA virus (*Figure 1—figure supplement 1A*). Moreover, GST-NEMO without its N-terminal IKK-binding site (NEMOΔN) rescued IRF3 activity in *Nemo$^{-/-}$(Ikbkg$^{-/-}$)* MEF cell extracts (*Figure 1—figure supplement 1B*), indicating an IKK-independent role of NEMO in virus-induced IRF3 activation. NEMO has been reported to interact with TBK1 through TANK (*Zhao et al., 2007*). Indeed, NEMOΔN pulled down endogenous TANK and TBK1 from *Nemo$^{-/-}$* cell extracts (NEMOΔN PD, *Figure 1—figure supplement 1C*). After NEMO depletion by a NEMO antibody, S100 lost its ability to support IRF3 dimerization in vitro, and the activity was restored by adding back NEMO PD, but not NEMO alone (*Figure 1C*). This suggests that NEMO and the TBK1 complex function together in IRF3 activation. However, NEMOΔN PD does not fully replace Q-B in IRF3 activation in vitro even in the presence of ubiquitin and E1 (data not shown), indicating that additional factor(s) might be required for IRF3 activation.

We further fractioned Q-B on Heparin-Sepharose and tested the ability of individual fractions to support IRF3 dimerization in the presence or absence of NEMOΔN PD. In this assay, we replaced Q-A with purified Ubc5 and also included purified ubiquitin and E1 to avoid identifying these known factors. Several fractions from the Heparin column showed IRF3 stimulatory activity, which was dependent on NEMOΔN PD (e.g., fraction 14 in *Figure 1—figure supplement 1D*). Subsequently, five more steps of conventional chromatography were used to purify this activity (*Figure 1—figure supplement 1E*). Fractions from the last monoQ column were subjected to silver staining and tandem mass spectrometry, which identified several proteins, including TRAF6. Immunoblotting with a TRAF6 antibody confirmed that TRAF6 co-purified with the IRF3 dimerization activity (*Figure 1—figure supplement 1F*).

### TRAF6, TRAF2, and TRAF5 are important for IRF3 and IKK activation in vitro

To determine whether TRAF6 is important for IRF3 activation in vitro, we performed reconstitution experiments using purified proteins and found that TRAF6 supported IRF3 activation in a manner that depended on MAVSΔTM and the ubiquitin system (*Figure 1D*). Similarly, IRF3 activation by virus-activated mitochondria (P5) was dependent on TRAF6 (*Figure 1—figure supplement 1G*). Cytosolic extracts from *Traf6$^{-/-}$* primary MEF cells were severely, albeit not completely, defective in supporting IRF3 dimerization and IκBα phosphorylation in vitro, and these defects were rescued by adding back wild-type TRAF6 (*Figure 1E,F*). In contrast, TRAF6 RING mutant (TRAF6-C70A), TRAF6 Zinc finger deletion (TRAF6ΔZF), or TRAF6 with the TRAF-C domain replaced by a fragment of bacterial gyrase-B (T6RZC) (*Wang et al., 2001*) failed to rescue IRF3 activation in *Traf6$^{-/-}$* cell extracts (*Figure 1E*). These results suggest that both TRAF6 E3 ligase activity and its ability to interact with other proteins, that is, MAVS (*Seth et al., 2005*; *Xu et al., 2005*), are important for IRF3 activation in vitro. However, it has been shown that *Traf6$^{-/-}$* cells exhibited normal virus-induced interferon production (*Seth et al., 2005*; *Zeng et al., 2009*). Consistent with these reports, *Traf6$^{-/-}$* primary MEF cells supported IRF3 and IKK activation in response to Sendai virus infection, but were defective in activating IKK in response to

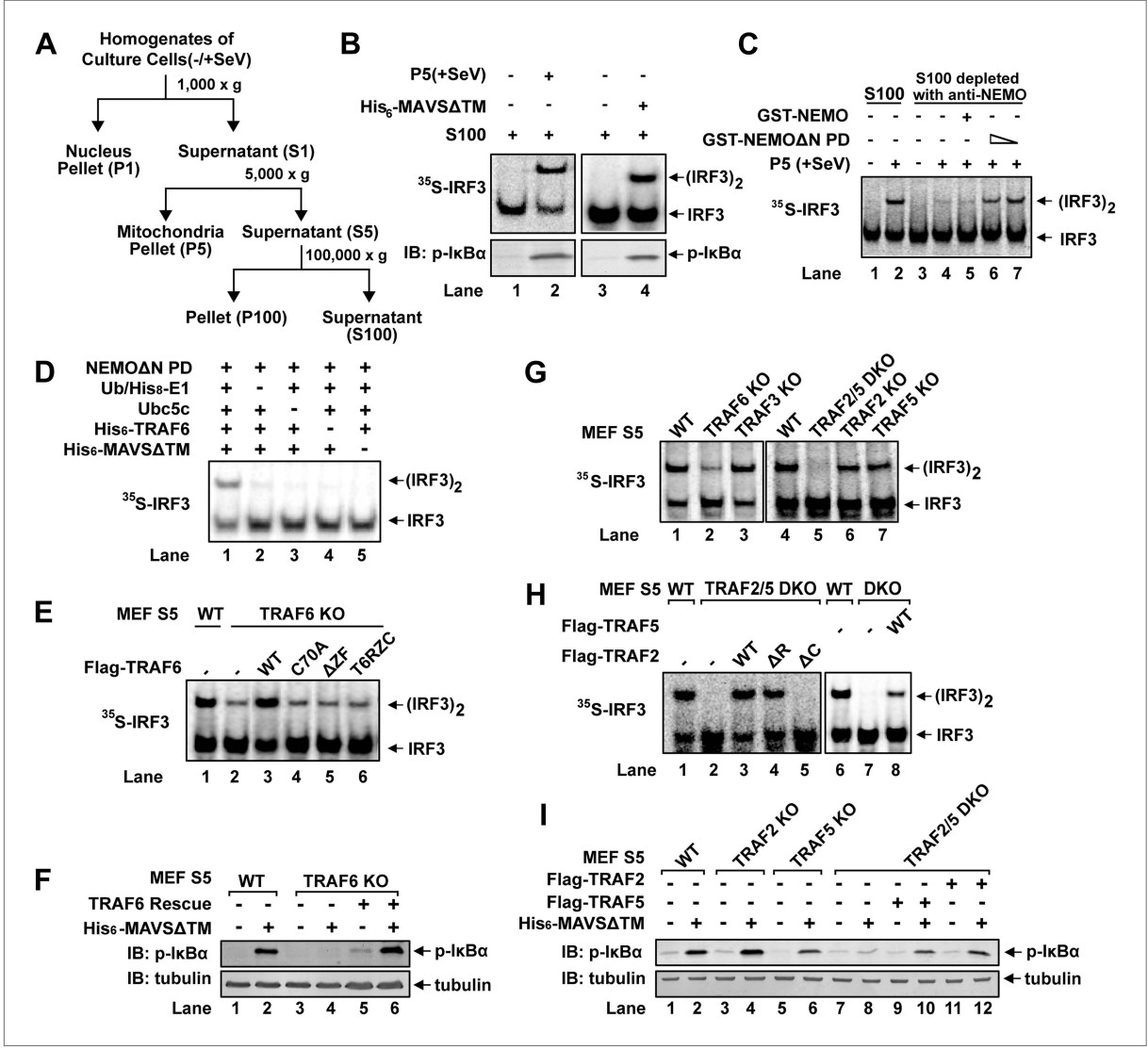

**Figure 1**. TRAF2, TRAF5, and TRAF6 are important for IRF3 and IKK activation in vitro. (**A**) Diagram of differential centrifugation of cell homogenates. HEK293T cells infected with Sendai virus (+SeV) or mock treated (−SeV) were homogenized in hypotonic buffer, followed by sequential centrifugation to separate crude mitochondria (P5) from cytosolic supernatant (S5 and S100). (**B**) IRF3 and NF-κB activation in vitro. Mitochondrial fraction (P5) from Sendai virus-infected HEK293T cells or purified His$_6$-tagged MAVS without transmembrane domain (His$_6$-MAVSΔTM) was incubated with cytosolic extract (S5) from uninfected cells in the presence of ATP and $^{35}$S-IRF3. Dimerization of IRF3 was analyzed by native gel electrophoresis, followed by autoradiography. IκBα phosphorylation was analyzed by immunoblotting. (**C**) NEMO-interacting complex is required for IRF3 activation in vitro. GST-tagged NEMO without its N-terminal IKK-binding region (GST-NEMOΔN) was mixed with cytosolic extract from *Nemo$^{−/−}$* MEF cells to collect GST-NEMOΔN pull down (NEMOΔN PD). This material, or GST-NEMO, was incubated with cytosolic extract (S100) from HeLa cells depleted of NEMO with a NEMO antibody. Activation of IRF3 was analyzed as described in (**B**). (**D**) Reconstitution of IRF3 dimerization in vitro. NEMOΔN PD, His$_8$-E1, Ubc5c, His$_6$-TRAF6, ubiquitin, His$_6$-MAVSΔTM, His$_8$-IRF3, and $^{35}$S-Flag-IRF3 were incubated together with ATP as indicated, followed by analysis of IRF3 dimerization. (**E** and **F**) TRAF6 is important for IRF3 and NF-κB activation by MAVS in vitro. Cytosolic extracts from wild-type or *Traf6$^{−/−}$* MEF cells were incubated with His$_6$-MAVSΔTM together with WT or mutant TRAF6 protein as indicated, followed by analysis of IRF3 dimerization (**E**) or IκBα phosphorylation (**F**). T6RZC: TRAF6 containing the RING, zinc, and coiled-coil domains, with the TRAF-C domain replaced by a fragment of bacterial gyrase B. (**G**) TRAF2 and 5 are important for IRF3 activation by MAVS in vitro. Cytosolic extracts from WT or different TRAF deficient MEF cells were incubated with His$_6$-MAVSΔTM, followed by IRF3 dimerization assay. (**H** and **I**) Either TRAF2 or TRAF5 rescues IRF3 and IKK activation by MAVS in the *Traf2/5* DKO extract. *Traf2/5* DKO extracts were supplemented with TRAF2 or TRAF5 proteins as indicated, together with His$_6$-MAVSΔTM and ATP, followed by measurement of IRF3 dimerization and IκBα phosphorylation.

The following figure supplements are available for figure 1:

**Figure supplement 1**. TRAF6 and TRAF2/5 are IRF3 activators.

IL-1β (*Figure 1—figure supplement 1H*). Thus, it is possible that other TRAF proteins might also be involved in MAVS signaling. Indeed, crude extracts (S5) from *Traf2⁻/⁻/Traf5⁻/⁻* (*Traf2/5* DKO) MEF cells failed to support IRF3 dimerization or IκBα phosphorylation in vitro; both activities were restored by adding back wild-type TRAF2 or TRAF5 (*Figure 1G,H,I*). Unlike TRAF6, TRAF2 RING deletion (TRAF2ΔR) did not impair IRF3 dimerization in cytosolic extracts (S5) from *Traf2/5* DKO cells. In contrast, deletion of the TRAF-C domain from TRAF2 (TRAF2ΔC) abolished its activity. MEF cells lacking a single TRAF protein, TRAF2, TRAF3, or TRAF5, were largely normal in activating IRF3 in vitro (*Figure 1G*).

## TRAF6 functions redundantly with TRAF2 and TRAF5 to activate IRF3 and IKK in cells

Similar to *Traf6⁻/⁻* MEFs, *Traf2⁻/⁻* MEFs induced IFNβ and IL6 normally in response to Sendai virus infection (*Figure 2—figure supplement 1A,B*). Moreover, in contrast to the profound defect of *Traf2/5* DKO cell extracts in supporting IRF3 and IKK activation by MAVS, the DKO cells activated IRF3 and IKK and induced IFNβ normally after Sendai virus infection (*Figure 2A,B*). Importantly, knockdown of TRAF6 expression by short hairpin RNA (shTRAF6) in the DKO cells abolished IRF3 and IKK activation and IFNβ induction by Sendai virus, but did not impair STAT1 phosphorylation induced by IFNγ (*Figure 2A*). The defects in IRF3 and IκBα phosphorylation in the DKO+shTRAF6 cells were rescued by expressing RNAi-resistant WT TRAF6, but not the C70A mutant of TRAF6. Similarly, VSV induction of several cytokines, including IFNβ, IL6, IFNα, and CXCL10, was abolished in the DKO+shTRAF6 cells but rescued by WT TRAF6 (*Figure 2C–E*, *Figure 2—figure supplement 1C,D*). A TRAF6 mutant in which every lysine was substituted with an arginine (T6-K0), but not TRAF6-C70A, rescued the cytokine expression in the DKO+shTRAF6 cells. Thus, TRAF6 functions redundantly with TRAF2 and TRAF5 in a manner that depends on the E3 ligase activity of TRAF6 but not its ubiquitination at any lysine residue.

Previous reports have suggested that TRAF3 and cIAPs function as ubiquitin E3 ligases that activate TBK1 and IRF3 (*Häcker et al., 2006*; *Oganesyan et al., 2006*). However, we have shown that in *Traf3*-deficient MEFs, RNA viruses can still activate IRF3 normally and induce IFNβ at a modestly reduced level (*Zeng et al., 2009*). To determine if TRAF3 functions redundantly with other TRAF proteins, we knocked down TRAF3 in *Traf2/Traf5* DKO MEFs and found that VSV induced IFNβ and IL6 normally in these cells (*Figure 2—figure supplement 1E–G*). Similarly, knocking down TRAF6 in *Traf3*-deficient MEFs did not impair IRF3 activation by Sendai virus (*Figure 2—figure supplement 1H*). To test the role of cIAPs in the MAVS pathway, we used a small molecule mimetic of SMAC, which is known to target the degradation of cIAP1 and cIAP2 (*Li et al., 2004*). This treatment (+SM) did not impair the activation of IRF3 or the induction of IFNβ or IL6 by VSV in WT, *Traf6⁻/⁻* or *Traf2⁻/⁻Traf5⁻/⁻* MEFs (*Figure 2—figure supplement 2A–D*). Taken together, these results suggest that TRAF3 and cIAPs are not essential for MAVS signaling, whereas TRAF6 functions redundantly with TRAF2 and TRAF5 to mediate MAVS signaling.

To test if the E3 ligase activity of TRAF2 is important for MAVS signaling, we expressed WT or a RING domain mutant (C34A) of TRAF2 in the DKO+shTRAF6 cells. In contrast to a requirement of the E3 ligase activity of TRAF6 in MAVS signaling (*Figure 2C–E*), both WT and C34A mutant of TRAF2 rescued virus activation of IRF3 as well as the induction of IFNβ, IL6, IFNα, and CXCL10 in these cells (*Figure 2F–H*, *Figure 2—figure supplement 2E,F*). WT and C34A TRAF2 also rescued IκBα phosphorylation induced by TNFα, but not IL-1β, in DKO+shTRAF6 cells (*Figure 2—figure supplement 2G*). These results suggest that the E3 ligase activity of TRAF2 is either dispensable or redundant with that of another E3 ligase (see below).

## The E3 ligase activities of LUBAC and TRAF2 function redundantly to support MAVS signaling

Recent studies suggest that the linear ubiquitin E3 complex LUBAC plays a positive role in NF-κB activation by TNFα but a negative role in regulating the RIG-I pathway (*Tokunaga et al., 2009*; *Inn et al., 2011*; *Belgnaoui et al., 2012*). To test the role of LUBAC in MAVS signaling, we used shRNA to knock down each individual subunit of the LUBAC complex, HOIP, HOIL-1, and Sharpin, as well as MAVS or TRAF3 in wild-type MEF cells (*Figure 3A,B*, *Figure 3—figure supplement 1A,B*). As expected, the knockdown of MAVS blocked IRF3 activation and IFNβ induction by VSV. The knockdown of HOIL-1 slightly enhanced IRF3 activation and IFNβ induction as compared to the control cells, consistent with its role as a negative regulator of RIG-I signaling (*Inn et al., 2011*). However, the knockdown of HOIP,

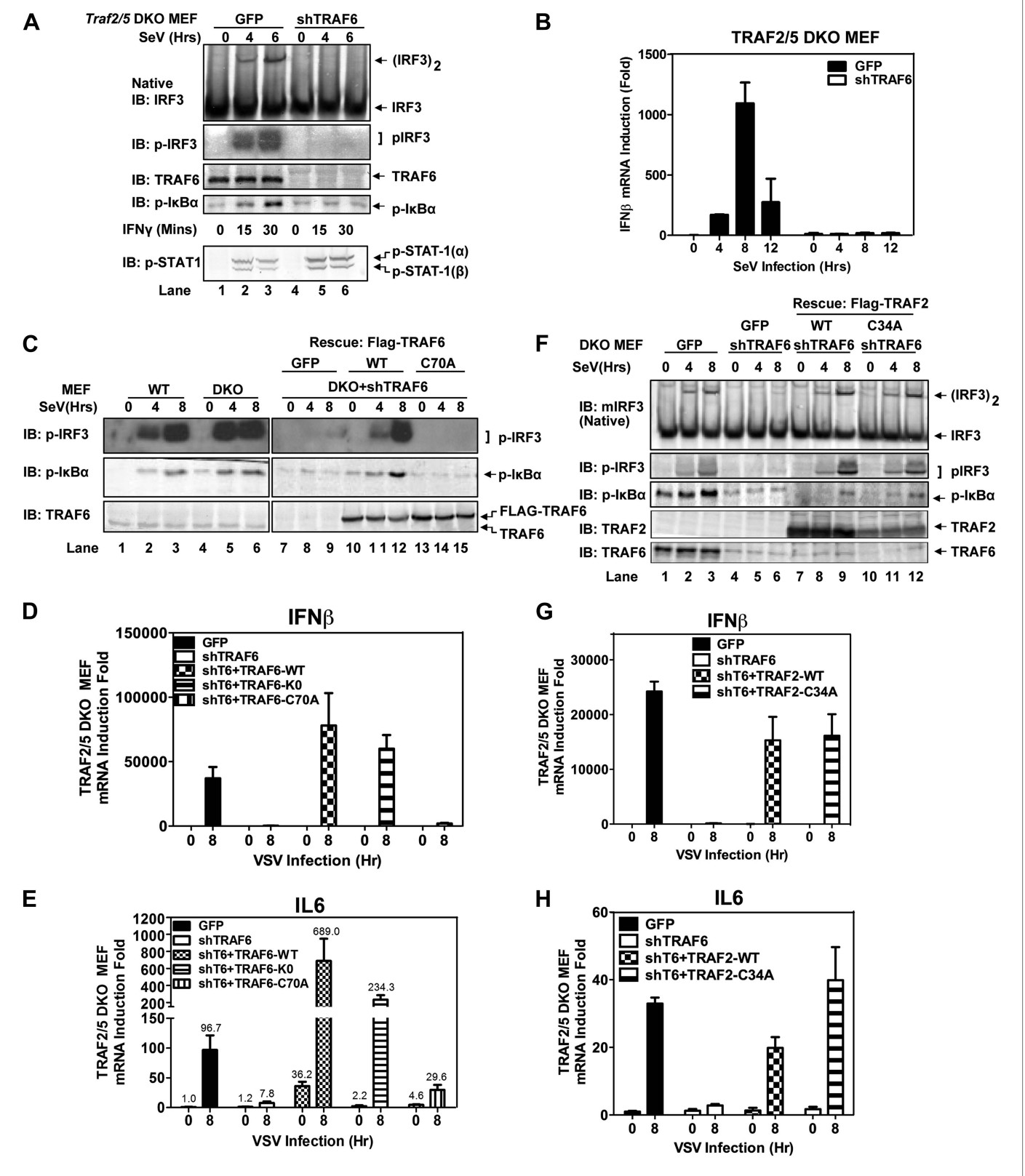

**Figure 2**. TRAF6 functions redundantly with TRAF2 and TRAF5 to activate IRF3 in cells. (**A**) Depletion of TRAF6 in *Traf2/5* DKO cells abolishes both IRF3 and NF-κB activation by virus. *Traf2/5* DKO MEF cells stably expressing GFP (as a control) or an shRNA against TRAF6 were infected with Sendai virus for the indicated time followed by immunoblotting of the cell extracts with the indicated antibodies (top). As a control, IFNγ induced STAT-1 phosphorylation

*Figure 2. Continued on next page*

*Figure 2. Continued*
was also analyzed by immunoblotting (bottom). (**B**) Depletion of TRAF6 in *Traf2/5* DKO cells abolishes IFNβ mRNA induction by virus. The cells described in (**A**) were treated with Sendai virus for the indicated time before total RNA was isolated. IFNβ mRNA level was analyzed by q-RT-PCR. (**C–E**) The catalytic activity of TRAF6 is required for antiviral immune responses. *Traf2/5* DKO MEF cells stably expressing shRNA against TRAF6 (DKO+shT6) and those in which endogenous TRAF6 was replaced with WT or RING mutant (C70A) Flag-TRAF6 were stimulated with Sendai Virus or VSV for the indicated time. Phosphorylation of IRF3 and IκBα was analyzed by immunoblotting. Total RNA was also isolated for the measurement of IFNβ and IL6 RNA by q-RT-PCR. T6-K0: a TRAF6 mutant in which all lysine residues were substituted with arginine. (**F–H**) The RING domain of TRAF2 is dispensable for its signaling functions. *Traf2/5* DKO cells stably expressing WT or RING mutant (C34A) Flag-TRAF2 were stimulated with Sendai virus or VSV for the indicated time. Activation of IRF3 and phosphorylation of IκBα was analyzed by immunoblotting. Cytokine RNA levels were measured by q-RT-PCR. Unless indicated otherwise, error bars in this and other figures of this paper represent standard deviations of triplicate experiments.
The following figure supplements are available for figure 2:
Figure supplement 1. TRAF2, 5, and 6 function redundantly to activate IRF3 in cells.
Figure supplement 2. Multiple E3 ligases function redundantly to activate IRF3 in cells.

Sharpin, or TRAF3 did not appreciably affect IRF3 activation and modestly inhibited IFNβ induction by VSV (*Figure 3A,B*). A previous report showed that IFNβ induction is enhanced in MEF cells derived from the mouse strain *Sharpin^cpdm*, which is deficient in Sharpin expression (*Belgnaoui et al., 2012*). Contrary to this report, we found that loss of Sharpin slightly reduced IFNβ induction by VSV (*Figure 3C*). At later time points, Sharpin deficiency partially reduced IFNβ induction by VSV and Sendai virus (*Figure 3—figure supplement 1C*). Re-introducing Sharpin back into *Sharpin*-deficient MEFs did not inhibit or enhance the activation of IRF3 or IKK by Sendai virus (*Figure 3—figure supplement 1D*). Consistent with our model that LUBAC is not a negative regulator of MAVS signaling, tetracycline-inducible knockdown of HOIP in the human osteosarcoma cell line U2OS modestly inhibited the activation of IRF3 and induction of IFNβ by VSV (*Figure 3—figure supplement 2A–C*).

To test if HOIP functions redundantly with TRAF2, we knocked down HOIP in MEF cells lacking TRAF2, TRAF5, and TRAF6 (*Traf2/5* DKO + shTRAF6), which were reconstituted with WT or ΔRING TRAF2 (*Figure 3D–F*) or WT TRAF6 (*Figure 3—figure supplement 2D*). Interestingly, the depletion of HOIP abolished IRF3 activation by VSV in the cells expressing TRAF2-ΔRING, but not in those expressing WT TRAF2 or TRAF6. The defect of IRF3 activation in the HOIP-depleted cells that expressed TRAF2-ΔRING was rescued by WT HOIP but not the catalytically inactive HOIP mutant, C693S/C696S (*Figure 3D*). Similar to HOIP, knockdown of Sharpin, and to a lesser extent HOIL-1, inhibited IRF3 and IKK activation in the *Traf2/Traf5* DKO + shTRAF6 cells expressing TRAF2 ΔRING (*Figure 3—figure supplement 2E*). Taken together, these results suggest that the E3 ligase activity of LUBAC can support MAVS signaling when the functions of the TRAF proteins are compromised.

## MAVS signals through its TRAF-binding motifs

MAVS harbors binding motifs for TRAF2, TRAF3, TRAF5, and TRAF6 (*Seth et al., 2005*; *Xu et al., 2005*; *Saha et al., 2006*; *Paz et al., 2011*) (*Figure 4A*). The motif PVQET (143–147) is known to bind TRAF2 and TRAF5, and perhaps TRAF3, whereas two motifs, PGENSE (153–158) and PEENEY (455–460) bind TRAF6. The second TRAF6-binding motif has also been suggested to bind TRAF3 (*Paz et al., 2011*). We showed previously that TRAF6 and TRAF2 shifted to high-molecular weight fractions together with MAVS polymers in response to Sendai virus infection (*Hou et al., 2011*). Here, we examined whether mutations in MAVS that selectively disrupt its binding to specific TRAF proteins affect the downstream signaling. *Mavs^−/−* MEFs stably expressing MAVS mutants were infected with Sendai virus followed by immunoblotting to examine IRF3 dimerization (*Figure 4B,C*) or q-RT-PCR to measure cytokine expression (*Figure 4D,E*). Mutations of the TRAF2/3/5 (Q145N) or TRAF6 [E155D/E457D (2ED)] binding sites did not affect IRF3 activation by Sendai virus (*Figure 4B*). In contrast, mutations of all known TRAF-binding sites (QN2ED) abolished IRF3 activation. Similarly, QN2ED, but not Q145N or 2ED, mutations in MAVS abolished IFNβ and IL-6 induction by Sendai virus. QN2ED MAVS also permitted higher levels of VSV replication (*Figure 4D–F*). To confirm the specificity of the TRAF-binding motifs in MAVS, endogenous MAVS was knocked down by shRNA in *Traf6^−/−* (*Figure 4G*) or *Traf2/5* DKO (*Figure 4H*) MEFs, which were then rescued with different RNAi-resistant MAVS mutants.

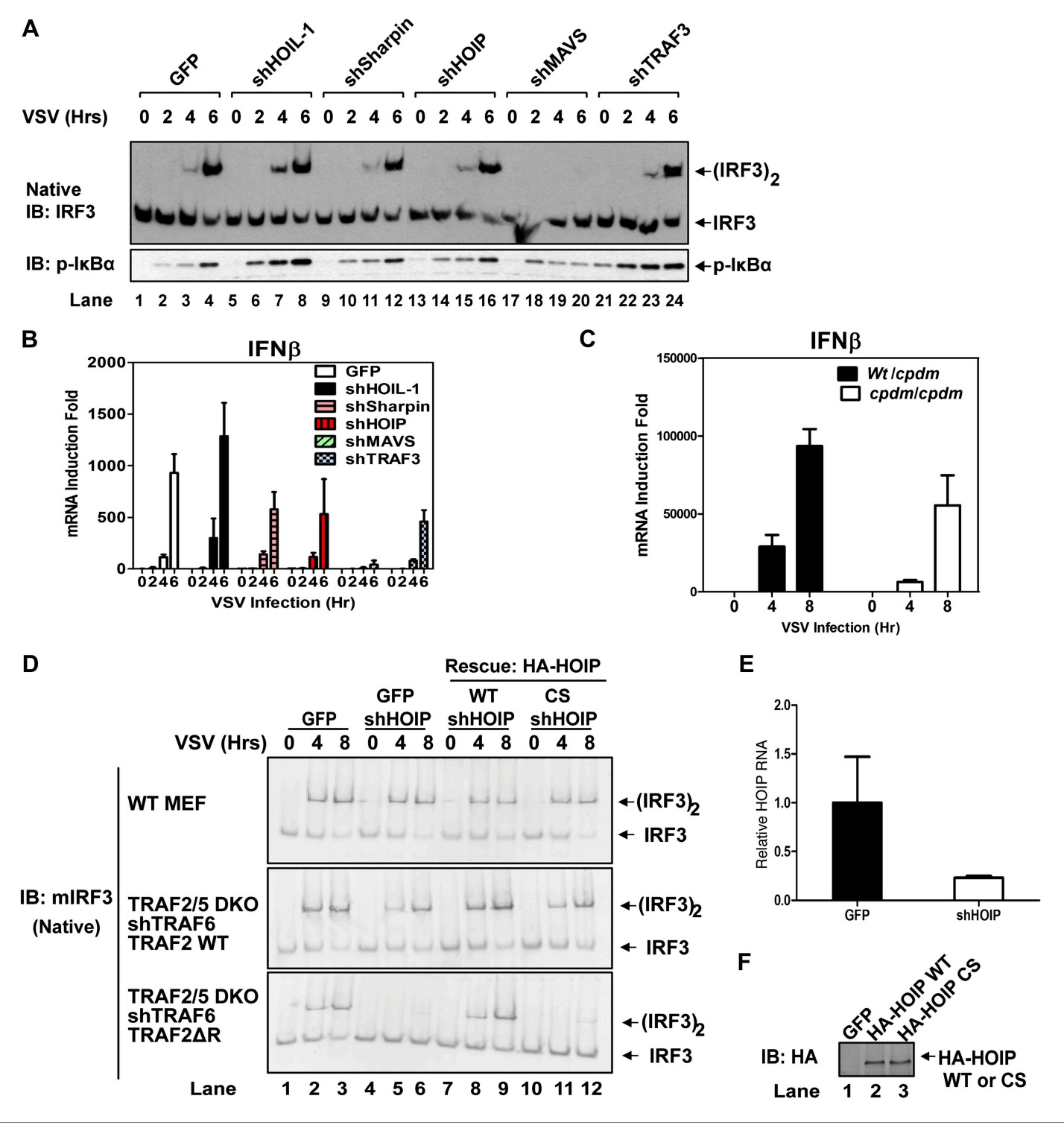

**Figure 3**. LUBAC functions redundantly with TRAF2 to support MAVS signaling. (**A** and **B**) WT MEF cells stably expressing GFP or shRNA against HOIL-1, Sharpin, HOIP, MAVS, or TRAF3 were infected with VSV for the indicated time. IRF3 dimerization and IκBα phosphorylation were analyzed by immunoblotting (**A**). The levels of IFNβ RNA were measured by q-RT-PCR (**B**). The efficiency of RNAi is shown in *Figure 3—figure supplement 1A,B*. (**C**) Primary MEF cells from heterozygous or *Sharpin^cpdm* mice were infected with VSV for the indicated time, and then IFNβ RNA levels were measured by q-RT-PCR. (**D**) *Traf2/5* DKO MEF cells stably expressing an shRNA against TRAF6 were reconstituted with TRAF2 WT or ΔRING (ΔR) mutant (lower panels). These cells, as well as WT MEF (upper panel), were further depleted of HOIP (lane 4–6) by lentiviral shRNA and then rescued with WT or the active site mutant (CS) of HOIP (lanes

*Figure 3. Continued on next page*

*Figure 3. Continued*

7–12). In lanes 1–3, a lentiviral vector expressing GFP was used as a control. The cells were infected with VSV for the indicated time, followed by measurement of IRF3 dimerization. (**E** and **F**) The cells described in (**D**) were analyzed for the expression of HOIP by q-RT-PCR (**E**) or immunoblotting with an HA antibody (**F**).

The following figure supplements are available for figure 3:

**Figure supplement 1**. LUBAC is largely dispensable for MAVS signaling when it is depleted from wild-type cells.

**Figure supplement 2**. LUBAC is required for MAVS signaling only when the functions of TRAF proteins are compromised.

These cells were complemented with either Flag-TRAF6 (*Figure 4G*) or Flag-TRAF2 (*Figure 4H*). Immunoprecipitation with Flag antibody revealed that the TRAF6 complex from the virus-infected cells contained WT and Q145N MAVS, but not 2ED or QN2ED MAVS (*Figure 4G*). In contrast, the TRAF2 complex contained WT and 2ED MAVS, but not Q145N or QN2ED MAVS (*Figure 4H*). Like WT MAVS, the QN2ED mutant formed high molecular weight aggregates after virus infection (*Figure 4—figure supplement 1A*), suggesting that its inability to activate IRF3 is likely due to defective recruitment of TRAF proteins rather than a defect in polymerization.

To further investigate the role of individual TRAF protein in MAVS signaling, we knocked down MAVS in MEF cells deficient in TRAF6, TRAF2, TRAF5, or both TRAF2 and TRAF5 (DKO), which were then rescued with RNAi-resistant WT or mutant MAVS (*Figure 4—supplement 1B–G*). In *Traf6⁻/⁻* cells, Q145N but not 2ED MAVS was completely defective in inducing IFNβ, but this defect was rescued by ectopic expression of WT but not C70A TRAF6 (*Figure 4—figure supplement 1B,C*). These results indicate that a functional TRAF6 was indispensable for signaling through the two TRAF6-binding sites (i.e., no other TRAF proteins, such as TRAF3, could substitute the function of TRAF6). In *Traf2/5* DKO cells, 2ED but not Q145N MAVS was defective in inducing IFNβ, but this defect was rescued by WT TRAF2 and partially rescued by the C34A and RING-deleted mutant of TRAF2 (TRAF2ΔRING) (*Figure 4—figure supplement 1D,E*). In *Traf2⁻/⁻* or *Traf5⁻/⁻* cells, only QN2ED MAVS failed to rescue IFNβ induction, suggesting that TRAF2 and TRAF5 could functionally substitute each other (*Figure 4—figure supplement 1F,G*). Similar to WT cells (*Figure 4B*), *Traf3⁻/⁻* cells could support IRF3 activation by WT, Q145N, and 2ED MAVS, but not QN2ED MAVS (*Figure 4—figure supplement 2A*). In contrast, *Traf6⁻/⁻* cells could not support IRF3 activation by Q145N MAVS. Taken together (*Figure 4—figure supplement 2B*), these results strongly support the conclusion that MAVS signals through its specific TRAF-binding motifs that recruit TRAF2, TRAF5, and TRAF6, which then function in parallel to activate IRF3 and NF-κB.

## MAVS polymerization is required for its recruitment of TRAF6

To test if MAVS polymerization is required for its binding to TRAF proteins, we mutated several conserved charged residues within MAVS CARD, including E26A, R64A, and R65A (*Figure 5A*). Each of these mutations abrogated the ability of MAVS to induce an IFNβ luciferase (IFNβ-Luc) reporter gene stably integrated in HEK293T cells (*Figure 5B*). WT or the mutant MAVS proteins were stably expressed in *Mavs⁻/⁻* MEFs, which were infected with Sendai virus followed by analysis of IRF3 dimerization. MAVS polymerization was analyzed by semidenaturing detergent agarose gel electrophoresis (SDD-AGE; *Figure 5C*; *Hou et al., 2011*). Unlike wild-type MAVS, which activated IRF3 and formed SDS-resistant aggregates following viral infection, MAVS containing point mutations in the CARD domain failed to activate IRF3 or form aggregates. Similarly, in HEK293T cells in which endogenous MAVS was knocked down by shRNA and replaced by RNAi-resistant MAVS WT or CARD mutants, only WT MAVS activated IRF3 after Sendai virus infection and formed high molecular weight particles that sedimented to the bottom layers after sucrose gradient ultracentrifugation (*Figure 5D* and *Figure 5—figure supplement 1A*). Importantly, each of the point mutations in MAVS CARD abolished the virus-induced interaction between TRAF6 and MAVS (*Figure 5E*). The mutations that disrupt MAVS polymerization also abrogated its binding to TRAF2 and TRAF5 (*Figure 5—figure supplement 1B,C*). These results strongly suggest that MAVS polymerization is important for the recruitment of TRAF6, TRAF2, TRAF5, and likely other signaling molecules.

## NEMO forms a complex with MAVS and TRAF proteins

Because the catalytic activity of TRAF6 is important for IRF3 activation in the MAVS pathway, and both K63 and linear polyubiquitination have been suggested to be important for IKK activation (*Deng et al., 2000*;

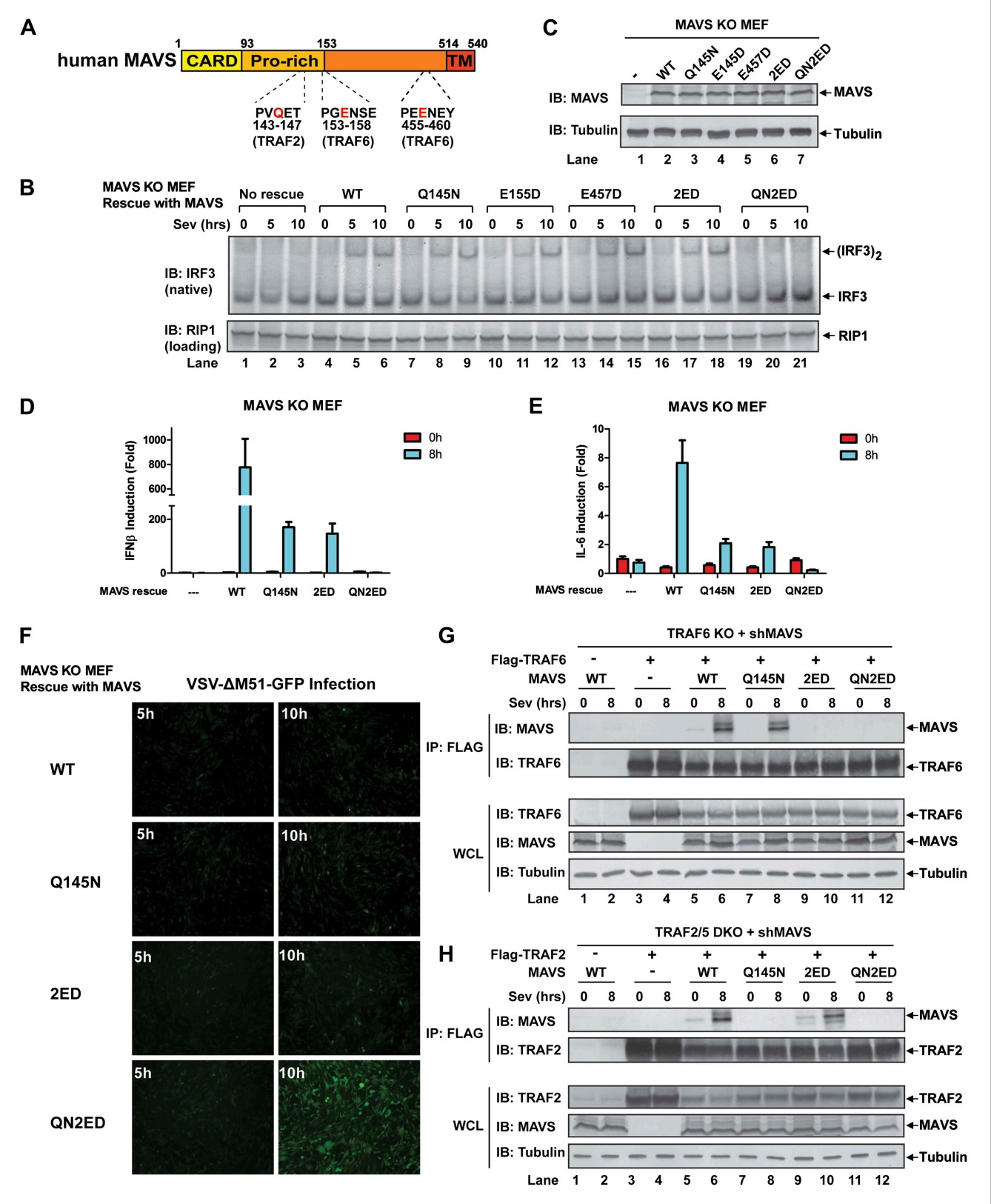

**Figure 4**. MAVS recruits multiple TRAF proteins to activate IRF3 upon virus infection. (**A**) The conserved binding motifs of TRAF2, TRAF5, and TRAF6 in human MAVS. Residues in red indicate mutation sites. (**B–E**) The TRAF-binding motifs on MAVS are essential for IRF3 and NF-κB activation by virus. *Mavs*[−/−] MEF cells reconstituted with WT or TRAF-binding mutant MAVS were infected with Sendai virus for the indicated time. IRF3 dimerization and the

*Figure 4. Continued on next page*

*Figure 4. Continued*

expression of different MAVS mutants were analyzed by immunoblotting. Cytokine RNA levels were measured by q-RT-PCR. 2ED: E155D/E457D; QN2ED: Q145N/E155D/E457D. (**F**) The TRAF-binding motifs on MAVS are important for restricting viral replication. MEF cells described in (**B**) were infected with VSV-ΔM51-GFP for the indicated time, and the GFP-positive cells were visualized under fluorescence microscope. (**G**) MAVS recruits TRAF6 after virus infection through TRAF6-binding motifs. *Traf6⁻/⁻* MEF cells were depleted of endogenous MAVS by shRNA, then reconstituted with Flag-TRAF6 and WT or mutant MAVS as indicated. The cells were infected with Sendai virus for the indicated time followed by immunoprecipitation of TRAF6 with a Flag antibody. The immunoprecipitates and whole cell lysates (WCL) were analyzed by immunoblotting with the indicated antibodies. (**H**) MAVS recruits TRAF2 upon virus infection through TRAF2 binding motif. Similar to (**G**), except that *Traf2/5 DKO* MEF cells were reconstituted with Flag-TRAF2 and the binding between TRAF2 and different MAVS mutants was analyzed.

The following figure supplements are available for figure 4:

**Figure supplement 1**. MAVS recruits multiple TRAF proteins upon virus infection.

**Figure supplement 2**. MAVS recruits multiple TRAF proteins upon virus infection.

*Wang et al., 2001*; *Haas et al., 2009*; *Tokunaga et al., 2009*; *Zeng et al., 2009*), we tested the effect of several ubiquitin mutants in IRF3 activation by MAVSΔTM in the in vitro assay (*Figure 6A*). We used a tetracycline-inducible RNAi system to knock down endogenous ubiquitin in the human cell line U2OS (*Xu et al., 2009*), and supplemented the cell extracts with different ubiquitin mutants, including K63-only, in which all lysine residues except K63 was substituted with arginine, His$_6$-Ub, which cannot form linear polyubiquitin chains, and K63R. The strongest defect in IRF3 dimerization and IκBα phosphorylation was observed in cell extracts containing K63R, suggesting that K63 polyubiquitination is important for IRF3 and IKK activation by MAVS in vitro.

NEMO contains two ubiquitin binding domains, a NEMO–ubiquitin binding (NUB; also known as UBAN) domain and a C-terminal zinc finger domain (*Ea et al., 2006*; *Wu et al., 2006*; *Laplantine et al., 2009*). We have shown previously that both of these domains are important for NEMO to mediate IRF3 activation in response to RNA virus infection (*Zeng et al., 2009*). To identify ubiquitination target(s) that relays upstream signal to NEMO, we incubated Flag-NEMO with HeLa S100 in the presence of MAVSΔTM. MAVS and TRAF2 were found to co-immunoprecipitate (Co-IP) with Flag-NEMO after incubation at 30°C but not at 0°C (*Figure 6B*). The MAVS-TRAF2-NEMO interaction was blocked when a deubiquitination enzyme, which contains the ovarian tumor type (OTU) domain of the Crimean Congo hemorrhagic fever virus, was included in the reaction mixture. Moreover, a NEMO mutant (UBDm; Y308S/H413A/C417A), which contained point mutations in both ubiquitin-binding domains, failed to pull down the TRAF2-MAVS complex (*Figure 6C*). These results suggest that MAVS and TRAF2 form a MAVS-induced ubiquitination-dependent signaling complex with NEMO in vitro.

To further characterize the MAVS signaling complex, we preformed two sets of stable isotope labeling by amino acids in cell culture (SILAC) experiments to identify proteins that associate with NEMO in a manner that depends on MAVS signaling and NEMO–ubiquitin binding (*Figure 6—figure supplement 1A,C*). In the first set (*Figure 6—figure supplement 1A*), extracts from MEF cells labeled with 'heavy' isotopes (K8R10) were stimulated with MAVSΔTM (A1 and B1) in the presence of WT Flag-NEMO, whereas extracts containing 'light' labeled proteins (K0R0) were either not stimulated with MAVSΔTM in the presence of WT Flag-NEMO (A2) or stimulated with MAVSΔTM in the presence of ubiquitin-binding defective NEMO mutant (Flag-NEMO-UBDm: Y308S/H413A/C417A; B2). After immunoprecipitation with the Flag antibody, the precipitates from the 'heavy' and 'light' samples were combined, fractionated by SDS-PAGE and analyzed by nano liquid chromatography tandem mass spectrometry (nanoLC-MS/MS). Among the proteins enriched in the heavy samples, we found that PLK1, TRAF2, and cIAP1 bound to NEMO in a manner that depended on MAVS and UBD of NEMO (*Figure 6—figure supplement 1B*). PLK1 was previously shown as a negative regulator of MAVS (*Vitour et al., 2009*). All three subunits of the LUBAC complex, HOIP, HOIL-1, and Sharpin, bound to NEMO in a UBD-dependent manner, but this binding was largely independent of stimulation by MAVS. MAVS itself was not identified as a hit because it was not labeled with the heavy isotopes.

In the second set of SILAC experiments (*Figure 6—figure supplement 1C*), we sought to identify proteins that associate with NEMO in living cells infected with VSV. We used *Nemo⁻/⁻* MEFs reconstituted with Flag-NEMOΔN, which lacks the IKK-binding site but remained capable of stimulating TBK1, or Flag-NEMOΔN-UBDm, which is defective in ubiquitin binding. The 'heavy' cells expressing

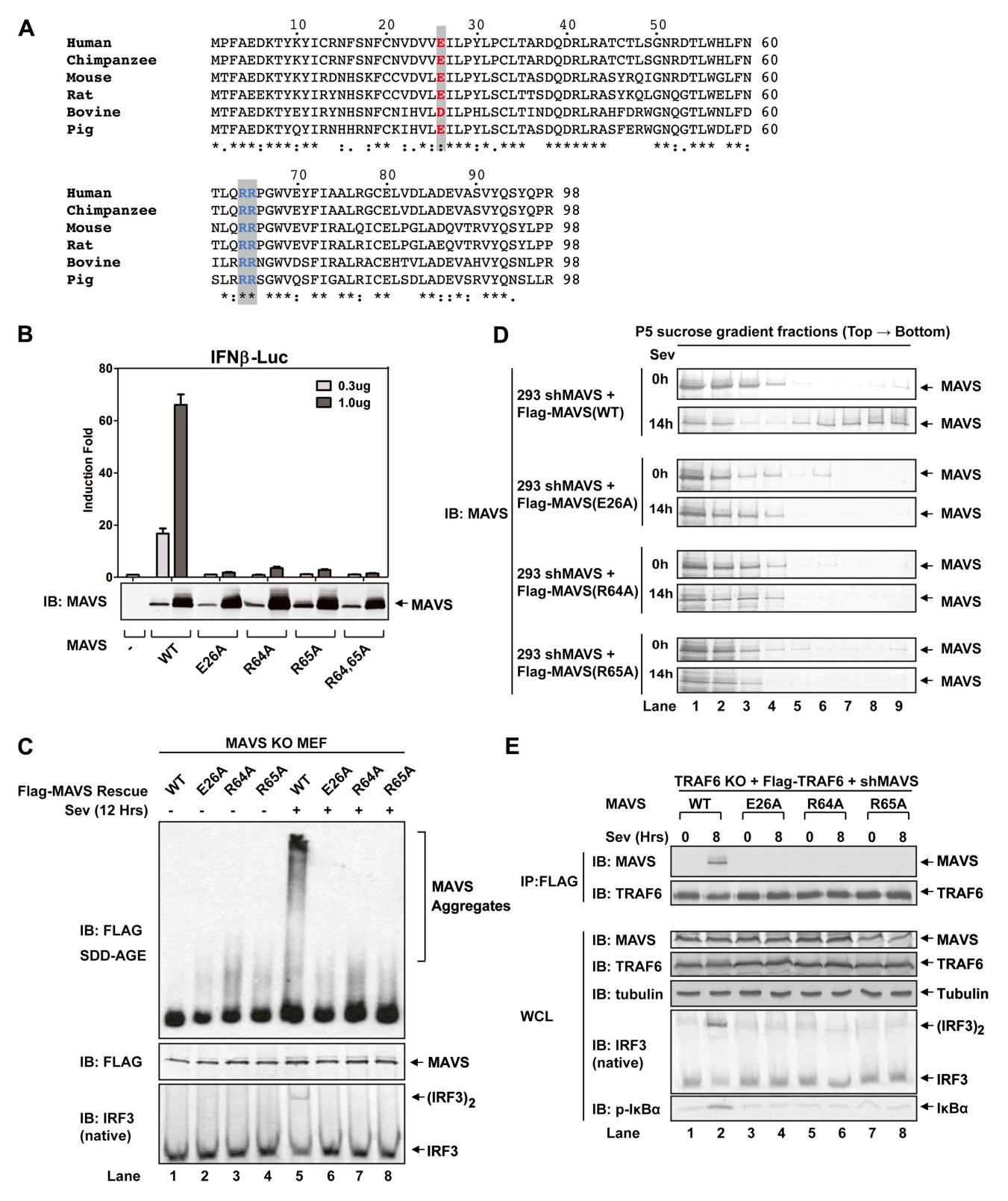

**Figure 5**. Prion-like polymerization of MAVS is required for IRF3 activation and TRAF6 recruitment. (**A**) Sequence alignment of the CARD domain of MAVS from different species. Conserved residues mutated in this study are colored and shaded. (**B**) MAVS WT or CARD mutants were transfected into HEK293-IFNβ-luciferase reporter cells. Cells were lysed 24 hr later, followed by luciferase reporter assay. (**C**) *Mavs^−/−* MEF cells reconstituted with
*Figure 5. Continued on next page*

*Figure 5. Continued*

Flag-MAVS WT or CARD mutants were infected with Sendai virus or mock treated for 12 hr, then mitochondrial extracts were separated by SDD-AGE (top) or SDS-PAGE (middle) followed by immunoblotting with a Flag antibody. Aliquots of the cytosolic extracts were separated by native gel electrophoresis (bottom), followed by immunoblotting with an IRF3 antibody. (**D**) HEK293 cells in which endogenous MAVS was knocked down by shRNA and replaced by RNAi resistant MAVS WT or CARD mutants were infected with Sendai virus for the indicated time. Crude mitochondria were solubilized in a buffer containing 1% DDM and then subjected to sucrose gradient ultracentrifugation. Aliquots of the fractions were immunoblotted with a MAVS antibody. (**E**) MAVS CARD mutants defective in polymerization failed to recruit TRAF6 upon virus infection. *Traf6⁻/⁻* MEF cells were depleted of MAVS by shRNA and reconstituted with Flag-TRAF6 and WT or mutant MAVS as indicated. The cells were infected with Sendai virus for the indicated time, and TRAF6 was then immunoprecipitated with a Flag antibody. Co-immunoprecipitated MAVS was analyzed by immunoblotting.

The following figure supplements are available for figure 5:

**Figure supplement 1**. Mutations that disrupt MAVS polymerization abolish viral activation of IRF3.

Flag-NEMOΔN (A1 and B1) and the 'light' cells expressing Flag-NEMOΔN-UBDm (B2) were infected with VSV, whereas the 'light' cells expressing Flag-NEMOΔN were uninfected (A2). Extracts from these cells were subjected to immunoprecipitation followed by nanoLC-MS/MS. Among the proteins that associated with NEMO WT, but not NEMO UBDm, after VSV infection (*Figure 6—figure supplement 1D*), several were known NEMO-interacting proteins, including TRAF6, TRAF3, TRAF2, A20, and MAVS. In addition, several proteins previously not known to associate with NEMO were identified. The role of these proteins in the MAVS pathway requires further investigation.

A comparison of the NEMO interacting proteins identified by the in vitro and cell-based SILAC experiments revealed only two common proteins, MAVS and TRAF2 (*Figure 6—figure supplement 1E*). At present, we do not know why some proteins such as TRAF6, TRAF3, and A20 interacted with NEMO only in the 'in vivo' but not the in vitro SILAC, whereas the LUBAC subunits behaved oppositely. One possibility is that these proteins interact with NEMO in a dynamic manner, which is recapitulated differently between the in vitro and 'in vivo' experiments. In any case, as TRAF2 was recruited to NEMO in a manner that depended on MAVS, VSV, and UBD of NEMO, we investigated whether ubiquitination of TRAF2 plays a role in MAVS signaling. We incubated Flag-tagged mouse TRAF2 with S100 from *Traf2/5* DKO MEFs in the presence of MAVSΔTM, followed by Flag immunoprecipitation under denaturing conditions (see 'Materials and methods'). Five ubiquitination sites on TRAF2 were detected by mass spectrometry, including K31, K148, K195, K313, and K481 as illustrated in *Figure 6D*. Among these lysines, K31 was identified as an autoubiquitination site of TRAF2 in the TNF pathway (*Li et al., 2009*). We mutated these lysine residues to arginine in TRAF2 (T2-5KR), and stably expressed the mutants as well as the WT TRAF2 in *Traf2/5* DKO+shTRAF6 MEFs. Both WT and 5KR TRAF2 rescued the induction of IFNβ, IL6, and CXCL10 by VSV (*Figure 6E–G*). Further mutation of another lysine (K38; to generate 6KR) also did not impair the function of TRAF2 (data not shown). Interestingly, knockdown of HOIP abolished the expression of these cytokines in cells expressing TRAF2-5KR, but not wild type TRAF2. These results, together with the results shown in *Figure 3*, suggest that ubiquitinated TRAF2 may function redundantly with another ubiquitination target of HOIP.

## Ubiquitin-binding by NEMO, but not ubiquitination of NEMO, is essential for MAVS signaling

NEMO has been reported to be ubiquitinated at lysine 285 and 309, and this modification is important for NF-κB activation (*Abbott et al., 2004*; *Tokunaga et al., 2009*). We found that *Nemo⁻/⁻* MEF cells stably expressing NEMO K285/309R mutant produced IFNβ and IL6 after VSV infection and that the cytokine levels were similar to those produced by cells expressing wild-type NEMO (*Figure 7A–C*). In contrast, the cells expressing NEMO-UBDm failed to induce the cytokines after VSV infection. To identify other potential ubiquitination sites on NEMO, we incubated endogenous amount of His₆-Flag-NEMO with cytosolic extracts from *Nemo⁻/⁻* MEF cells in the presence of MAVSΔTM. NEMO was purified under denaturing conditions and analyzed by nanoLC-MS/MS (see 'Materials and methods'), which detected signal-dependent ubiquitination of NEMO at K285, K325, K342, and K344 (*Figure 7D*). However, *Nemo⁻/⁻* MEF cells stably expressing NEMO 6KR (K111/285/309/325/342/344R) mutant still supported virus-induced cytokine production similar to those expressing wild-type NEMO (*Figure 7E,F*). We also constructed a NEMO mutant in which 13 conserved lysine residues were substituted with arginine (13KR) and found that this mutant could still support IRF3 dimerization in *Nemo*-deficient cell

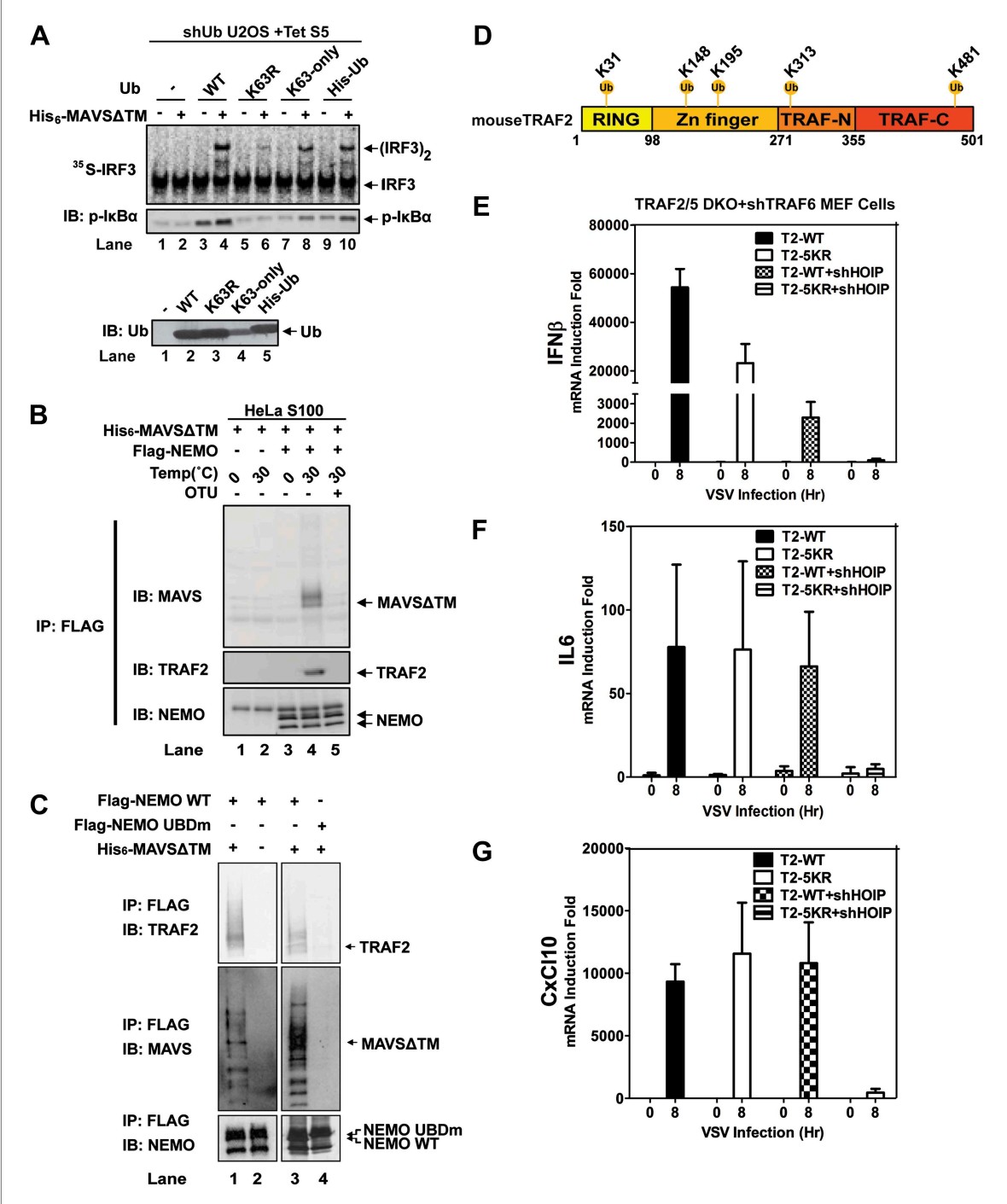

**Figure 6**. Ubiquitination-dependent assembly of a MAVS signaling complex. (**A**) Lys63-linked polyubiquitination is important for MAVS-mediated IRF3 and IKK activation. U2OS cells stably integrated with tetracycline-inducible shRNA against ubiquitin genes were grown in the presence of tetracycline for 48 hr to deplete endogenous ubiquitin. The cytosolic extracts were then supplemented with WT or mutant ubiquitin (1 µg) and $^{35}$S-IRF3 in the presence or absence of MAVSΔTM, followed by analyses of IRF3 dimerization and IκBα phosphorylation. K63-only: containing only one lysine at residue 63 of ubiquitin. Ubiquitin WT and mutants were analyzed by immunoblotting with an ubiquitin antibody. (**B**) Purified Flag-NEMO was incubated with HeLa S100 and His$_6$-MAVSΔTM at the indicated temperatures in the presence or absence of vOTU, a viral deubiquitination enzyme. After the reaction, NEMO was immunoprecipitated with Flag antibody, and the co-precipitated proteins were detected with specific antibodies. (**C**) Similar to (**B**), except that Flag-NEMO WT and UBD mutant were tested for their ability to interact with TRAF2 and MAVS. (**D**) A schematic diagram of TRAF2 protein. The ubiquitination sites identified by mass spectrometry are highlighted. (**E–G**) *Traf2/5* DKO MEF cells stably expressing shRNA against TRAF6 were reconstituted with WT

*Figure 6. Continued on next page*

*Figure 6. Continued*

TRAF2 or a TRAF2 mutant containing arginine at each of the five ubiquitination sites (K31/148/195/313/481R, 5KR). In some experiments, HOIP was further knocked down by shRNA as indicated. These cells were infected with VSV for the indicated time and cytokine RNA levels were analyzed by q-RT-PCR.
The following figure supplements are available for figure 6:

**Figure supplement 1**. SILAC experiments to identify ubiquitin-dependent NEMO-signaling complex both in vitro and in cells.

extracts in the presence of MAVSΔTM (*Figure 7G,H*). These results suggest that NEMO ubiquitination per se is dispensable for IRF3 activation in the MAVS pathway. It is likely that ubiquitination of multiple targets as well as unanchored K63 polyubiquitin chains function cooperatively to recruit NEMO and other signaling molecules to the MAVS polymers and promote the activation of IKK and TBK1 (*Figure 8* and 'Discussion').

## Discussion

Although it is known that MAVS contains binding sites for several TRAF proteins (*Seth et al., 2005*; *Xu et al., 2005*), the role of these TRAF proteins in MAVS signaling has been enigmatic because cells lacking an individual TRAF protein, including TRAF2, TRAF3, TRAF5, and TRAF6, could still induce IFNβ normally in response to virus infection (*Seth et al., 2005*; *Konno et al., 2009*; *Zeng et al., 2009*). In this report, we demonstrated that compound deletion of TRAF6, TRAF2, and TRAF5 completely abolished IFNβ induction by Sendai virus and VSV, suggesting that these proteins function redundantly in vivo to activate the downstream signaling cascades after they are recruited to the cognate binding sites on MAVS. Such redundant function of the TRAF proteins was not recapitulated in our cell free assay, in which IRF3 activation by MAVS was strongly inhibited in *Traf*6-deficient extracts, and largely abolished in the absence of TRAF2 and TRAF5. It is possible that certain aspects of cell signaling in intact cells, such as the potential involvement of cellular structures (e.g., membrane organelles) and transcriptional amplification, are not recapitulated in the in vitro system that employs soluble cytosolic extracts. Nevertheless, the in vitro assay recapitulates many aspects of MAVS signaling, such as its dependency on MAVS and NEMO. In the assay that involves endogenous MAVS, IRF3 activation occurs only when cytosolic extracts are incubated with mitochondria from virus-infected cells. Thus, the in vitro assays that we have established are valuable tools to dissect the biochemical mechanisms of MAVS signaling. Our finding that MAVS recruits multiple ubiquitin E3 ligases and other signaling proteins to activate IKK and TBK1 further underscores the importance of combining in vitro biochemical approach with cell-based assays to gain a mechanistic understanding of these complex signaling pathways.

Currently, the prevailing view is that TRAF2, TRAF5, and TRAF6 mediate the activation of IKK and NF-κB, whereas TRAF3 is important for the activation of TBK1 and IRF3. However, here we showed that TRAF2, TRAF5, and TRAF6 are essential for the activation of both IKK and TBK1 by MAVS, whereas TRAF3 is dispensable. Paz et al. have recently shown that the TRAF6-binding site (455-PEENEY-460) of MAVS also binds to TRAF3 and that this binding is important for MAVS signaling (*Paz et al., 2011*). Another recent report confirmed that the PEENEY motif (residues 455–460) of MAVS is important for IFNβ induction when an intermediate domain of MAVS is fused to an artificial oligomerization domain (*Takamatsu et al., 2013*). However, this study did not directly investigate whether TRAF6 or TRAF3 is important for MAVS signaling. Our data showed that in cells expressing a MAVS mutant lacking the TRAF2/5 binding sites (Q145N), activation of IRF3 and production of IFNβ were completely abolished in cells depleted of TRAF6, indicating that other TRAF proteins, including TRAF3, could not substitute the function of TRAF6. It is possible that the role of TRAF3 in IRF3 activation depends on the cell types and specific signaling pathways. Alternatively, TRAF3 may regulate type-I interferon production through a mechanism that does not affect IRF3 activation.

Our finding that TRAF2, TRAF5, and TRAF6 play an essential role in both NF-κB and IRF3 activation by MAVS is surprising because these TRAF proteins are also recruited to other receptors such as TNF receptor, TLRs, IL-1 receptor, and CD40, which only stimulate NF-κB but not IRF3. A possible explanation for these observations is that IRF3 activation requires certain adaptor proteins such as MAVS, STING, and TRIF. Indeed, we have recently shown that STING simultaneously binds to TBK1 and IRF3, thereby specifying IRF3 phosphorylation by TBK1 (*Tanaka and Chen, 2012*). This mechanism explains why

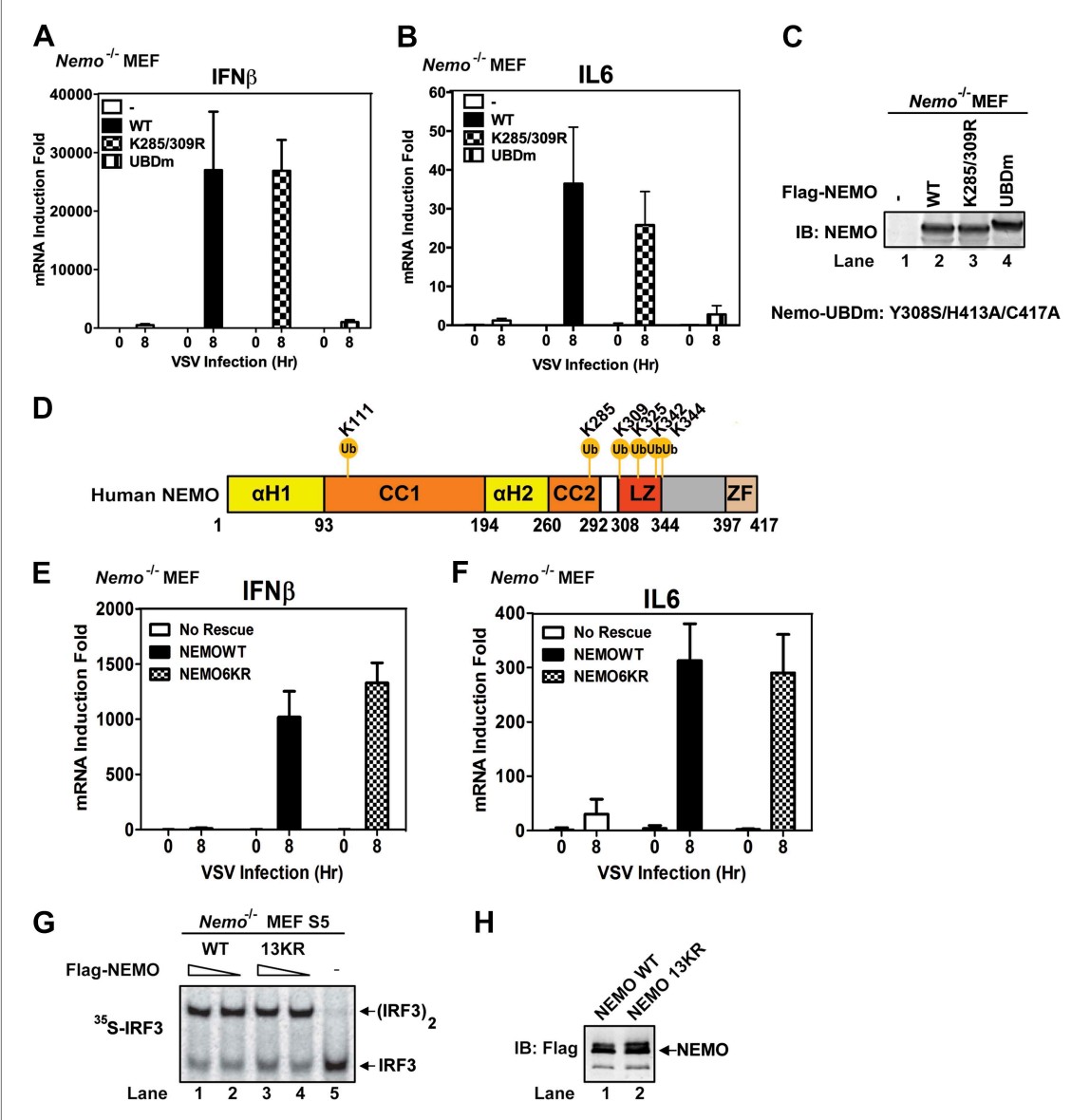

**Figure 7**. Mutation of NEMO ubiquitination sites does not impair viral induction of interferons. (**A–C**) *Nemo⁻/⁻* MEF cells stably expressing GFP, Flag-NEMO WT or mutants were infected with VSV for the indicated time. Cytokine RNA levels were measured by q-RT-PCR (**A** and **B**). Expression of the NEMO proteins was analyzed by immunoblotting (**C**). (**D**) A schematic diagram of human NEMO and the ubiquitination sites identified by mass spectrometry. (**E** and **F**) *Nemo⁻/⁻* MEF cells stably expressing GFP, Flag-NEMO WT, or 6KR (K111/285/309/325/342/344R) were infected with VSV for the indicated time. Cytokine RNA levels were analyzed by q-RT-PCR. (**G** and **H**) *Nemo⁻/⁻* MEF cell extracts (S5) were supplemented with Flag-NEMO WT or 13KR (K111/139/143/165/283/285/321/325/326/342/344/399R) protein and incubated with ³⁵S-IRF3 and His₆-MAVSΔTM. IRF3 dimerization was detected by native gel electrophoresis followed by autoradiography (**G**). Aliquots of the NEMO proteins were analyzed by immunoblotting (**H**).

several innate immunity pathways, such as the TLR and IL-1 pathways, could lead to activation of TBK1 but not IRF3 (**Clark et al., 2011**).

Our model that MAVS signals through multiple TRAF proteins is further supported by mutagenesis experiments which showed that mutations of both TRAF2/5 and TRAF6 binding sites of MAVS, but not each alone, abolished IRF3 and IKK activation by virus infection. Importantly, MAVS binds to the TRAF proteins in a manner that depends on virus infection and MAVS polymerization. Mutations in the CARD domain of MAVS that disrupt its polymerization also abolish the recruitment of the TRAF proteins. These results explain why MAVS undergoes prion-like polymerization in response to viral infections. Presumably, each TRAF binding motif of MAVS has a low affinity for the TRAF proteins, but

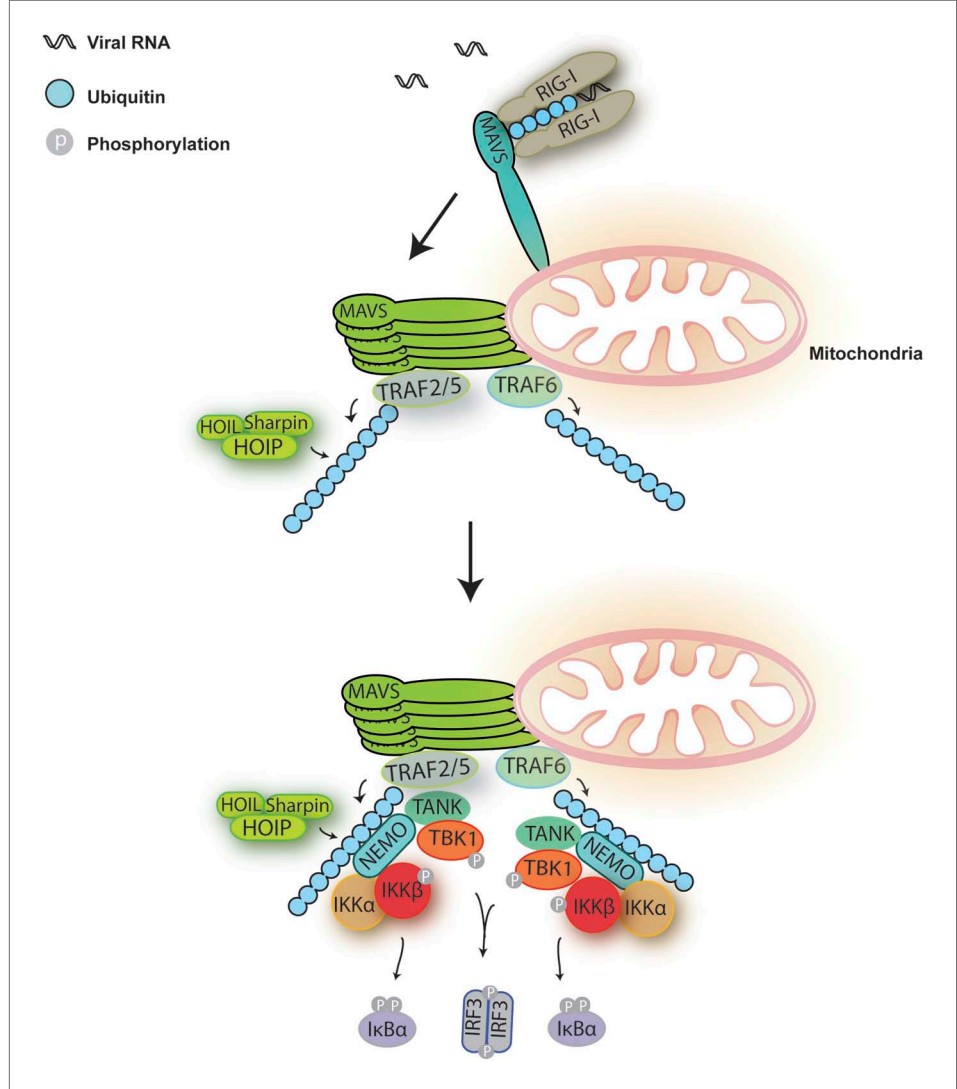

**Figure 8**. A Model for MAVS-mediated IRF3 and NF-κB activation. Upon virus infection, MAVS undergoes prion-like polymerization to recruit and activate E3 ligases TRAF2, TRAF5, and TRAF6 (possibly also LUBAC). These E3 ligases in turn synthesize polyubiquitin chains on TRAF2 and other proteins, resulting in the recruitment of NEMO through its ubiquitin-binding domains. NEMO then recruits IKK and TBK1 complexes to the MAVS polymer, where the kinases phosphorylate IκBα and IRF3, respectively, leading to the induction of type-I interferons and other cytokines.

the polymerization of MAVS significantly increases the avidity of the interaction, resulting in the recruitment of the TRAF proteins. The recruitment of TRAFs to the MAVS polymers may lead to the oligomerization of TRAFs, which activates their ubiquitin E3 ligase activity (*Wang et al., 2001*; *Sun et al., 2004*). The E3 ligase activity of TRAF6 is likely important for IRF3 activation because mutation of the RING domain in TRAF6 abolishes its activity. Interestingly, the RING domain of TRAF2 is important for IRF3 activation only when the catalytic activity of HOIP is abrogated, suggesting that TRAF2 and HOIP may have redundant functions. HOIP is dispensable for IRF3 activation in cells expressing TRAF2, TRAF5, and TRAF6. Conversely, the loss of TRAF2, TRAF5, and TRAF6 completely abolishes IRF3 activation in cells expressing other E3s such as LUBAC, cIAPs, and TRAF3. These results indicate that TRAF2, TRAF5, and TRAF6 are the major E3s that mediate the activation of IKK and TBK1 downstream of MAVS.

How do TRAF2, TRAF5, and TRAF6 activate IKK and TBK1? Both IKK and TBK1 employ NEMO as an essential regulatory subunit for their activation. NEMO contains two ubiquitin-binding domains, which are essential for the activation of IKK and TBK1 in response to virus infection. Although we could detect ubiquitination of NEMO at multiple lysines, mutations of these lysines to arginine did not impair

the ability of NEMO to support IRF3 activation by MAVS. In fact, a NEMO mutant containing mutations at 13 conserved lysine residues (13KR) could still activate IRF3 in the presence of MAVS (*Figure 7G,H*). We also found that TRAF2 is recruited to NEMO in a manner that depends on ubiquitination and MAVS. TRAF2 is also ubiquitinated at multiple lysine residues. However, mutations of these residues to arginine did not impair its ability to induce cytokines in response to VSV infection unless HOIP was also depleted (*Figure 6E–G*). Thus, HOIP may have an additional ubiquitination target that functions redundantly with TRAF2 ubiquitination to mediate IRF3 activation. An emerging theme from this work is that multiple ubiquitin E3s are recruited to MAVS, and each E3 targets ubiquitination of distinct proteins, which function cooperatively and redundantly to activate IKK and TBK1. A similar model has recently been proposed for the regulation of DNA repair by sumoylation (*Psakhye and Jentsch, 2012*). It was shown that sumoylation of a group of DNA repair proteins ('protein group'), rather than a specific protein target, is responsible for the repair of DNA double-strand breaks.

## Materials and methods

### Antibodies

Rabbit antibodies against human IRF3, TRAF2, TRAF3, TRAF6, NEMO, and mouse antibodies against human MAVS and ubiquitin were obtained from Santa Cruz Biotechnology, Dallas, TX; Rabbit antibody against human HOIP was from Abcam, Cambridge, MA; Flag antibody (M2), M2-conjugated agarose, anti-HA-conjugated agarose, and tubulin antibody were purchased from Sigma-Aldrich, St. Louis, MO; HA antibody was from Covance, Princeton, NJ; antibodies against pIRF3 Ser$^{396}$, pTBK1 Ser$^{172}$, pIκBα Ser$^{32/36}$, and pSTAT-1 Tyr$^{701}$ were from Cell Signaling, Danvers, MA; mouse IRF3 antibody was from Invitrogen, Carlsbad, CA; pan cIAP antibody was from R&D Systems, Minneapolis, MN; TBK1 antibody was from IMGENEX Corp, San Diego, CA; TANK antibody was from BioVision, Milpitas, CA. The antibody against mouse Sharpin was generated by immunizing rabbits with the full-length mouse Sharpin produced in Sf9 cells. The antibody against mouse HOIL-1 was kindly provided by Dr Kazuhiro Iwai (Kyoto University). Rabbit antibodies against human and mouse MAVS were generated as described before (*Seth et al., 2005*; *Sun et al., 2006*).

### Expression constructs and recombinant proteins

For expression in mammalian cells, mouse cDNA encoding N-terminal Flag or HA tagged TRAF6 WT, TRAF6 C70A, TRAF6 K0, TRAF2 WT, TRAF2 C34A, TRAF2ΔR (Δ34-72), TRAF2ΔC (1-359), TRAF2 K31R, TRAF2 5KR (K31/148/195/313/481R), TRAF2 6KR (K31/38/148/195/313/481R), and TRAF5 WT were cloned into pcDNA3. Human cDNA encoding N-terminal Flag or HA-tagged NEMO WT, K285/309R, 6KR (K111/285/309/325/342/344R), UBDm (Y308S/H413A/C417A), and ΔN (86-419) were cloned into pcDNA3 and pTY-EF1A-puroR-2a lenti-viral vectors. Human MAVS WT, QN (Q145N), 2ED (E155D and E457D), and QN2ED (Q145N, E155D and E457D) were cloned into pTY-EF1A-puroR-2a lenti-viral vector. Mouse HA-tagged HOIP WT and CS (C693S/C696S) mutant were cloned into pTY-EF1A-hygromycinR lentiviral vector (see below). Mutants were constructed with the QuikChange Site-Directed Mutagenesis Kit (Stratagene, La Jolla, CA). Flag or HA-tagged TRAF proteins were overexpressed in HEK293T cells, whereas Flag or HA-tagged NEMO proteins were overexpressed in *Nemo*$^{-/-}$ MEF cells. Proteins were purified with M2 or anti-HA agarose, followed by Flag or HA peptide elution. For expression in *Escherichia coli*, cDNA encoding N-terminal His$_6$-Flag-tagged NEMO WT or K285/309R was inserted into pET23a; cDNA encoding N-terminal His$_6$-tagged ubiquitin WT, K63R, K63 only, MAVSΔTM (aa1–510 or 1–460) was also cloned into pET23a. Vectors encoding ubiquitin mutants were transformed and expressed in *E. coli* BL21(DE3)-pJY2 strain to prevent misincorporation of Lys residues. Other vectors were transformed and expressed in *E. coli* BL21(DE3)-pLysS strain. His$_6$-MAVSΔTM (1–510 or 1–460) was purified in the presence of 4 M urea, whereas other His$_6$-tagged proteins were purified under native condition, by nickel affinity chromatography. Ubc5c, His$_6$-TRAF6, His$_8$-E1, and His$_8$-IRF3 were purified as described previously (*Zeng et al., 2009*; *Hou et al., 2011*). Nontagged ubiquitin and mutants were from Boston Biochem, Cambridge, MA.

### Viruses, cell culture, and transfections

Sendai virus (Cantell strain; Charles River Laboratories) was used at a final concentration of 100 hemagglutinating unit/ml. VSV (ΔM51)-GFP virus was from Dr John Bell (University of Ottawa) and propagated in Vero cells. All cells were cultured at 37°C in an atmosphere of 5% (vol/vol) $CO_2$. HEK 293T cells were

cultured in Dulbecco's modified Eagle's medium (DMEM) supplemented with 10% (vol/vol) cosmic calf serum (Hyclone, Thermo Fisher Scientific, Waltham, MA) with penicillin (100 U/ml) and streptomycin (100 μg/ml). *Traf2/5* DKO MEF cells and derivatives were cultured in DMEM supplemented with 20% (vol/vol) fetal bovine serum (Invitrogen) and antibiotics. Other MEF cells were cultured in DMEM supplemented with 10% (vol/vol) fetal bovine serum (Atlanta) and antibiotics. *Mavs*$^{-/-}$ MEFs were immortalized from *Mavs*$^{-/-}$ mice generated in **Sun et al. 2006**. Primary and immortalized *Traf6*$^{-/-}$ MEFs were generated from *Traf6*$^{-/-}$ mice provided by Dr Tak Mak (University of Toronto). *Traf2*$^{-/-}$ MEFs, *Traf5*$^{-/-}$ MEFs and *Traf2*$^{-/-}$/*Traf5*$^{-/-}$ MEFs were kindly provided by Dr Hiroyasu Nakano (Juntendo University School of Medicine). *Traf3*$^{-/-}$ MEFs were gifts from Dr Genhong Cheng (University of California, Los Angeles). *Nemo*-deficient MEFs and IKKα/IKKβ-deficient MEFs were kindly provided by Dr Inder Verma (Salk Institute); these cells are a complete null for the Ikbkg, Ikbkα and Ikbkβ loci, respectively. *Sharpin*-deficient MEFs were immortalized from *Sharpin*$^{cpdm}$ mice purchased from the Jackson Laboratory (Stock number: 007599).

## Purification of NEMO–TBK1 complex

*Nemo*$^{-/-}$ MEFs and MEFs stably expressing Flag-NEMOΔN were lysed in hypotonic buffer (10 mM Tris–HCl (pH 7.5), 10 mM KCl, 1.5 mM MgCl$_2$, and a protease inhibitor cocktail [Roche]). After centrifugation at 100,000×*g* for 30 min, the supernatants from both types of cells were mixed at a ratio of 5:1, and the mixture was subjected to immunoprecipitation with M2 agarose at 4°C overnight. The agarose beads were washed three times with buffer B (10 mM Tris–HCl [pH 7.5], 1 M NaCl, and 0.1% CHAPS), and the proteins were eluted with Flag peptide (0.2 mg/ml) in buffer C (50 mM Tris–HCl [pH 7.5], 0.1% CHAPS). The eluted proteins containing endogenous TBK1 and TANK from MEFs, designated as NEMOΔN PD were stored in buffer D (20 mM Tris–HCl [pH 7.5], 50 mM NaCl, 10% glycerol, and 0.1% CHAPS) after buffer exchange by repeated dilution and concentration.

## Biochemical assays for IRF3 activation, IκBα phosphorylation, and NEMO–MAVS complex formation

In vitro assays for IRF3 activation and IκBα phosphorylation were preformed as described previously (**Zeng et al., 2009**), except that 20 ng of His$_6$-MAVSΔTM was used to replace crude mitochondria (P5) from virus-infected cells in some experiments. In the assays for purification of IRF3 activators, the reaction mixture (10 μl) contained 100 ng His$_8$-E1, 50 ng Ubc5c, 5 μg ubiquitin, 20 ng His$_8$-IRF3, and 1 μl of NEMOΔN PD.

To determine the NEMO–MAVS complex formation in vitro, reaction mixture (100 μl) containing 20 mM HEPES-KOH (pH 7.0), 2 mM ATP, 5 mM MgCl$_2$, 200 ng His$_6$-MAVSΔTM, 200 μg cytosolic extracts (S5 or S100), 200 ng Flag-NEMO was incubated at 30°C for 1 hr. Immunoprecipitation was then carried out using Flag antibody (M2) agarose at 4°C overnight in the presence of 20 mM Tris–HCl (PH 7.5), 100 mM NaCl, 0.5% NP-40, and the protease inhibitor cocktail (Roche). The agarose beads were washed three times with lysis buffer A (20 mM Tris–HCl [PH 7.5], 100 mM NaCl, 10% glycerol, and 0.5% NP-40), and coprecipitated proteins detected by immunoblotting.

## Biochemical fractionation of cytosolic extract and purification of TRAF6

Protein purification was carried out using AKTA-FPLC or ETTAN system (GE Healthcare, Little Chalfont, UK) at 4°C. Chromatographic columns and media were purchased from GE Healthcare unless indicated otherwise. HeLa S100 was prepared as described previously from 50 L of cells purchased from National Cell culture Center (**Deng et al., 2000**). S100 was loaded onto a 60-ml Q-Sepharose column equilibrated with buffer Q-A (20 mM Tris–HCl [pH 7.5], 10% glycerol, and 0.02% CHAPS), and eluted with 0.25 M NaCl. The eluate was diluted in buffer SP-A (20 mM HEPES-KOH [pH 6.5], 10% glycerol, and 0.02% CHAPS) and concentrated repeatedly to reduce the salt before loading to SP-Sepharose. The flow-through from the SP column was further fractionated on a Heparin-Sepharose column with a linear gradient of NaCl (0–300 mM) in buffer SP-A, and active fractions eluted around 150 mM NaCl were pooled. After the salt was reduced by repeated dilution in buffer SP-A, the sample was fractionated on 2 ml ceramic hydroxyapatite (CHT) column (Bio-Rad Laboratories Inc., Hercules, CA) with a linear gradient of KH$_2$PO$_4$–K$_2$HPO$_4$ (0–300 mM) in buffer CHT-A (5 mM KH$_2$PO$_4$–K$_2$HPO$_4$, pH 7.0, 150 mM NaCl). Active fractions eluted around 150 mM PO$_4$ were pooled and precipitated with 30% ammonium sulfate. The pellet was resuspended with buffer Q-A and fractioned on a 2.4-ml Superdex 200 PC 3.2/30 column in buffer Q-A containing 100 mM NaCl. Active fractions containing proteins with a size of ~300 KD were loaded onto MonoQ-Sepharose and eluted with a linear gradient

of NaCl (150–350 mM) in buffer Q-A. The fractions were resolved by SDS-PAGE, silver stained, and identified by tandem mass spectrometry using LTQ-XL (Thermo).

## Lentiviral-mediated RNAi and rescue with transgenes

The lentiviral shRNA vector, pTY-shRNA-EF1a-puroR-2a-GFP-Flag, was provided by Dr Yi Zhang (Harvard Medical School). This vector was modified to pTY-shRNA-EF1a-hygroR-GFP and pTY-shRNA-EF1a-zeroR-GFP by replacing the puromycin resistant gene with hygromycin- and zeocin-resistant genes, respectively, such that multiple lentiviral vectors can be introduced to the same cell line. The original vector was also modified to pTY-shRNA-EF1a-GFP-IRES-puroR to circumvent the problem of incomplete cleavage by the 2A protease. The shRNA sequences were cloned into the vectors with U6 promoter. RNAi-resistant cDNA sequences were cloned into the vectors to replace GFP. Lentiviral infection and establishment of stable cells were described previously (*Tanaka and Chen, 2012*). The shRNA sequences are as follows (only the sense strand is shown): mouse TRAF6, 5′-GGATGATACATTACTAGTG-3′; mouse HOIP, 5′-GGCGCTCAGTGAAGTTTAA-3′; mouse MAVS, 5′-GATCAAGTGACTCGAGTTT-3′; human MAVS, 5′-GGAGAGAATTCAGAGCAAG-3′; mouse TRAF3, 5′-GAATGAAAGTGTTGAGAAA-3′; mouse HOIL-1, 5′-GGAGAAAGCCCGAGCTGTA-3′ (3′UTR); mouse Sharpin, 5′-GCTACATACAAGCTAGTAA-3′ (3′UTR); human HOIP, 5′-CCTAGAACCTGATCTTGCA-3′.

## MAVS aggregation and recruitment of TRAF proteins

MAVS aggregation induced by Sendai virus infection was analyzed by sucrose gradient ultracentrifugation as previously described (*Hou et al., 2011*). Briefly, crude mitochondria fraction was collected from the cells, loaded onto the top of 20–60% sucrose, and centrifuged at 170,000×*g* for 2 hr. Immunoblotting was used to analyze the distribution of proteins along the sucrose gradient. The formation of prion-like aggregates of MAVS was also analyzed by semidenaturing detergent agarose gel electrophoresis (SDD-AGE; see *Hou et al., 2011*).

To examine the recruitment of TRAF proteins to MAVS, cells were lysed with buffer containing 20 mM Tris–HCl (pH 7.5), 150 mM NaCl, 0.5% NP-40, and protease inhibitor cocktail (Roche, Basel, Switzerland). Flag antibody (M2) agarose was added to the cell lysates to immunoprecipitate TRAF proteins at 4°C overnight. The agarose beads were washed for five times with lysis buffer before being boiled in SDS loading buffer. Co-immunoprecipitated MAVS was then analyzed by immunoblotting.

## Quantitative reverse transcription PCR (q-RT-PCR)

Total RNA was isolated using TRIzol (Invitrogen). 0.1 µg total RNA was reverse-transcribed into cDNA using iScript Kit (Bio-Rad). The resulting cDNA served as the template for Quantitative-PCR analysis using iTaq Universal SYBR Green Supermix (Bio-Rad) and Vii™7 Real-Time PCR System (Applied Biosystems Inc., Foster City, CA). IQ SYBR Green Supermix (Bio-Rad) and iQ5 real-time PCR detection system (Bio-Rad) were used for some of the experiments. Primers for specific genes are listed as follows: Mouse β-actin, 5′-TGACGTTGACATCCGTAAAGACC-3′ and 5′-AAGGGTGTAAAACG CAGCTCA-3′; Mouse IFNβ, 5′-CCCTATGGAGATGACGGAGA-3′ and 5′-CTGTCTGCTGGTGGAGTTCA-3′; Mouse IFNα, 5′-ATTTTGGATTCCCCTTGGAG-3′ and 5′-TATGTCCTCACAGCCAGCAG-3′; Mouse CxCl-10, 5′-GGTCTGAGTGGGACTCAAGG-3′ and 5′-GTGGCAATGATCTCAACACG-3′; Mouse IL-6. 5′-TCCATCCAGTTGCCTTCTTG-3′ and 5′-GGTCTGTTGGGAGTGGTATC-3′; Mouse TRAF3, 5′-AGCAGC TGACTCTGGGACAT-3′ and 5′-CACCACACAGGGACAATCTG-3′; Mouse TRAF6, 5′-GCCCAGGCTGTT CATAATGT-3′ and 5′-CGGATCTGATGGTCCTGTCT-3′; Mouse HOIP, 5′-TTATGCGAGACCCCAAGTTC-3′ and 5′-GCCTTGAGCCTGGTACTCTG-3′. human IFNβ, 5′-ACTGCAACCTTTCGAAGCCTTT-3′ and 5′-TGGAGAAGCACAACAGGAGAGC-3′; human GAPDH, 5′-ATGACATCAAGAAGGTGGTG-3′ and 5′-CATACCAGGAAATGAGCTTG-3′.

## Purification of TRAF2 and NEMO for mapping posttranslational modifications

To determine modification(s) on TRAF2 induced by MAVSΔTM in cell extracts, reaction mixture (10 ml) containing 20 mM HEPES-KOH (pH 7.0), 2 mM ATP, 5 mM MgCl₂, 20 µg His₆-MAVSΔTM, 20 mg *Traf2/5* DKO cytosolic extracts (S100), and 20 µg Flag-TRAF2 was incubated at 30°C for 1 hr, followed by addition of 0.7% SDS to terminate the reaction. The reaction mixture was then diluted by adding 60 ml PBS, and immunoprecipitation was carried out using Flag antibody (M2) agarose at 4°C overnight in the presence of 0.5% NP-40. The agarose beads were washed three times with PBS containing 0.5% NP-40, and the proteins were eluted with Flag peptide before SDS-PAGE and silver staining.

To determine NEMO modification(s) induced by MAVSΔTM in cell extracts, reaction mixture (5 ml) containing 20 mM HEPES-KOH (pH 7.0), 2 mM ATP, 5 mM $MgCl_2$, 10 µg $His_6$-MAVSΔTM, 10 mg $Nemo^{-/-}$ cytosolic extracts (S100), and 10 µg $His_6$-Flag-NEMO was incubated at 30°C for 1 hr, followed by addition of 1% SDS to terminate the reaction. The reaction mixture was then diluted by adding 45 ml PBS and immunoprecipitation was carried out using Flag antibody (M2) agarose at 4°C overnight in the presence of 0.5% NP-40. The agarose beads were washed three times with PBS and 0.5% NP-40, and proteins were eluted with a buffer containing 20 mM Tris–HCl (PH 8.0), 100 mM NaCl, and 8M urea. The eluate was then incubated with Ni-NTA agarose at 4°C overnight. After washing three times with a buffer containing 20 mM Tris–HCl (pH 8.0), 100 mM NaCl, and 10 mM imidazole, the agarose beads were boiled in the SDS loading buffer before SDS-PAGE and silver staining.

For both TRAF2 and NEMO, gel slices from each lane were excised and digested with trypsin in situ. Digested samples were subjected to mass spectrometry using Q Exactive (Thermo Scientific), and raw data were analyzed by mascot search engine (MATRIX SCIENCE).

## SILAC experiments

Wild-type MEF and $Nemo^{-/-}$ MEF expressing Flag-NEMOΔN WT or UBDm were cultured in SILAC-DMEM medium lacking lysine and arginine. The medium was supplemented with dialyzed FBS, penicillin, streptomycin, and amino acids L-lysine and L-arginine. The 'light' culture was supplemented with Lys0 ($^{12}C_6^{14}N_2$) and Arg0 ($^{12}C_6^{14}N_4$), and the 'heavy' culture with Lys8 ($^{13}C_6^{15}N_2$) and Arg10 ($^{13}C_6^{15}N_4$). All SILAC reagents were purchased from Pierce (Thermo Scientific). The in vitro and cell-based SILAC experiments were preformed according to the outline described in *Figure 6—figure supplement 1A,C*, respectively. After SDS-PAGE and silver staining, 10 to 12 gel slices from each lane were excised and digested with trypsin in situ. Extracted peptides were fractionated on a homemade analytical column (75 µm ID, 100 mm in length) packed with C18 resin (100 Å, 3 µm, MICHROM Bioresources) using Dionex Ultimate 3000 nanoLC system (Thermo Scientific). The column was coupled in-line to a Q Exactive mass spectrometer (Thermo Scientific) equipped with a nano-electrospray ion source, which was set at a spray voltage of 2.3 kV. Peptides were eluted with a 78 min gradient as follows: 2–30% B in 68 min, 30–35% B in 4 min, 35–40% B in 2 min, 40–60% B in 3 min, and 60–80% B in 1 min (A = 0.1% formic acid; B = 100% acetonitrile in 0.1% formic acid). Full scan mass spectra were acquired from $m/z$ 300 to 1500 with a resolution of 70,000 at $m/z$ = 200 in the Orbitrap. MS/MS spectra (resolution: 17,500 at $m/z$ = 200) were acquired in a data-dependent mode, whereby the top 10 most abundant parent ions were subjected to further fragmentation by higher energy collision dissociation (HCD). SILAC data were processed using MaxQuant computational platform (*Cox and Mann, 2008*) version 1.3.0.5 which incorporates the Andromeda search engine (*Cox et al., 2011*). Proteins were identified by searching the mouse UniProt database supplemented with frequently observed contaminants. The first search tolerance was set at 20 ppm, and main search deviation at 6 ppm. The required minimum peptide length was six amino acids. The false discovery rate (FDR) at both peptide and protein levels was set to 0.01. SILAC quantification of each protein group was based on at least two ratio counts.

## Acknowledgements

We thank Dr Ming Xu for the U2OS ubiquitin replacement cells and Fenghe Du for assistance in molecular cloning.

## Additional information

### Competing interests

ZJC: Reviewing editor, *eLife.* The other authors declare that no competing interests exist.

### Funding

| Funder | Grant reference number | Author |
|---|---|---|
| National Institutes of Health | GM63692 | Siqi Liu, Jiaxi Wu, Zhijian J Chen |

| Funder | Grant reference number | Author |
|---|---|---|
| Cancer Prevention and Research Institute of Texas | RP101496 | Jueqi Chen |
| Welch Foundation | I-1389 | Xin Cai, Zhijian J Chen |
| Howard Hughes Medical Institute | | Xiang Chen, Lijun Sun, Zhijian J Chen |
| Cancer Prevention and Research Institute of Texas | RP110430 | You-Tong Wu, Zhijian J Chen |

The funders had no role in study design, data collection and interpretation, or the decision to submit the work for publication.

## Author contributions

SL, JC, Conception and design, Acquisition of data, Analysis and interpretation of data, Drafting or revising the article; XC, JW, XC, Acquisition of data, Analysis and interpretation of data; Y-TW, Acquisition of data, Contributed unpublished essential data or reagents; LS, Conception and design, Analysis and interpretation of data; ZJC, Conception and design, Analysis and interpretation of data, Drafting or revising the article

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
