## [Decision Letter]

Thank you for sending your work entitled “MAVS Recruits Multiple Ubiquitin E3 Ligases to Activate Antiviral Signaling Cascades” for consideration at *eLife*. Your article has been favorably evaluated by a Senior editor and 3 reviewers, one of whom is a member of our Board of Reviewing Editors.

The Reviewing editor and the other reviewers discussed their comments before we reached this decision, and the Reviewing editor has assembled the following comments to help you prepare a revised submission.

The manuscript by Liu et al. elucidated the biochemical mechanism for MAVS activation of type I interferon response and NF-κB signaling. Upon innate immune detection of cytoplasmic viral RNA by RIG-I receptors, MAVS, a mitochondrial outer membrane protein, polymerizes and signals NF-κB and IRF3 activation through IKK and TBK1 kinases. This process critically requires ubiquitination, but the nature of the ubiquitination events and the roles of TRAF-family ubiquitin ligases have been controversial and are not clearly defined. In this study, the authors employed cell-free reconstitution and biochemical fraction approaches, and identified TRAF6 being one of the key ubiquitin ligases in mediating MAVS activation of both NF-κB and IRF3 signaling. Using viral infection of different TRAF KO MEFs, the authors further discovered that TRAF2/5, but not TRAF3, play a redundant role with TRAF6 and function cooperatively with TRAF6 to mediate ubiquitination signaling upon MAVS activation. Importantly, all three TRAFs (TRAF2/5/6) are recruited to MAVS upon viral RNA-induced MAVS polymerization, and only when all three TRAFs are depleted is MAVS-mediated antiviral signaling completely blocked. This clarifies the role of different TRAF ubiquitin ligases in MAVS-mediated antiviral signaling. Moreover, the study also demonstrates that K63-linked ubiquitin chains, sensed simultaneously by the two ubiquitin-binding domains in NEMO, play a pre-dominant role in MAVS activation of NF-κB and IRF3 signaling through IKK/TBK1. These findings are of great interest and certainly seem to advance our understanding of the RIG-I-MAVS antiviral signaling. The study is carefully designed and performed, the evidences provided are quite thorough and complete, and the conclusion drawn is reasonably justified. However, there are several issues that need to be addressed before the manuscript can be accepted for publication:

1) The finding of LUBAC coordinating with TRAF2/5 in MAVS signaling contradicts with two prior reports that instead show a negative role (Inn et al. Mol. Cell 41:354; Belgnaoui et al. Cell Host & Microbe 12:211). The authors should try resolving this discrepancy. The authors should compare the impact of deficiency of each of the three LUBAC components on MAVS signaling as other studies suggested that, although the three proteins associate, they are differentially involved in distinct functions. The authors may also test this using different cell lines.

2) TRAF2/5 and LUBAC function redundantly in generating ubiquitin chain signals in the absence of TRAF6. Meanwhile, the authors also indicate that K63 ubiquitin chains play a major role in MAVS signaling. TRAF2/5 is suggested in some previous studies to have no E3 activity while LUBAC only produces linear chains. Therefore, the authors should address how K63 ubiquitin chains are generated in the absence of TRAF6. Do the authors really mean that TRAF2/5 and/or LUBAC can make K63 ubiquitin chains? The authors should also perform knockdown experiments to clearly rule out the role of cIAPs in *Traf6*-deficient cells.

3) The present study disagrees with several prior studies which suggested an essential role of TRAF3 in MAVS signaling. This is rather puzzling as the evidence presented previously was obtained independently by several groups. The authors should better investigate this discrepancy. Is this due to clonal variation and adaption among the TRAF KO cells? Are all different TRAFs expressed in KO cells as expected? The authors should also properly refer to those prior studies so that the contradiction can be more evident. For example, the paper does not give accurate reference to Oganesyan et al. (Nature 439:208). That paper is only referred as that TRAF3 is required for TLR-mediated IFN induction, though it did show that VSV activation of IRF3-dependent genes is markedly reduced in *Traf*3 KO MEFs.

4) Another major concern is the large quantitative differences among the experiments: The levels of IL6 induction attained in Figure 2 are in the range of 1000-fold, and in 2H of 60-fold. The levels of IFNβ induction in Figure 2 approaches 100,000-fold while those in Figure 2–figure supplement 1E are less than 600-fold. The author should clarify if these huge differences reflect just experimental variations or differences intrinsic to the cell clones used. The extent of the inter-clonal variation of response should be assessed by comparing a number of different cell clones of the same genotype. The authors should also clarify whether this large variation in extent of induction reflects differences in actual level of gene expression attained after viral infection or variation in the basal expression levels of the induced gene. In the latter case, the actual values of expression, rather than 'fold induction' should be presented. The exact nature of the differences in the values of gene induction in the various cells used in the same experiment is also not clear. Do the figures present average and standard deviation? In that case, average of what: different cell clones or several replicates of the same clone? In how many independent experiments? Are the various transfected proteins expressed at the same level and how do they compare to the endogenous levels? Western analyses and comparison of the expression levels of the transfected proteins should be presented.

---

## [Author Response]

*1) The finding of LUBAC coordinating with TRAF2/5 in MAVS signaling contradicts with two prior reports that instead show a negative role (Inn et al. Mol. Cell 41:354; Belgnaoui et al. Cell Host & Microbe 12:211). The authors should try resolving this discrepancy. The authors should compare the impact of deficiency of each of the three LUBAC components on MAVS signaling as other studies suggested that, although the three proteins associate, they are differentially involved in distinct functions. The authors may also test this using different cell lines*.

Following the reviewers’ suggestion, we have compared the effect of knocking down each of the three LUBAC components on MAVS signaling (new Figure 3, Figure 3–figure supplement 1). While knockdown of HOIL-1 led to a modest enhancement of IRF3 activation and IFN? induction, knockdown of HOIP or Sharpin in WT MEF cells did not affect IRF3 activation and only modestly inhibited IFN? induction. The positive role of HOIP and Sharpin in IRF3 activation was revealed only in cells expressing TRAF2?RING, but not WT TRAF2 (Figure 3; Figure 3–figure supplement 1I), supporting our model that HOIP and Sharpin function redundantly with the E3 activity of TRAF2. Overall, our results are consistent with those of Inn et al. that HOIL-1 is a negative regulator of RIG-I signaling (Mol Cell 41:354), but contradict with those of Belgnaoui et al. who showed that Sharpin negatively regulates the RIG-I pathway (Cell Host & Microbe 12:211). To further investigate the role of Sharpin in the RIG-I pathway, we infected the *Sharpin*-deficient (cpdm) MEFs with VSV and Sendai virus (SeV) for different time points (Figure 3–figure supplement 1C). In each case, IFN? induction by the viruses was significantly reduced in the absence of Sharpin. We also used tetracycline-inducible RNAi to knock down HOIP in U2OS cells and found that IFN? induction by VSV was partially inhibited in the absence of HOIP (Figure 3–figure supplement 1E–1G). Thus, in multiple independent experiments involving different cell lines, viruses and time courses, we consistently found that HOIP and Sharpin played a positive, albeit not indispensable, role in the RIG-I pathway.

*2) TRAF2/5 and LUBAC function redundantly in generating ubiquitin chain signals in the absence of TRAF6. Meanwhile, the authors also indicate that K63 ubiquitin chains play a major role in MAVS signaling. TRAF2/5 is suggested in some previous studies to have no E3 activity while LUBAC only produces linear chains. Therefore, the authors should address how K63 ubiquitin chains are generated in the absence of TRAF6. Do the authors really mean that TRAF2/5 and/or LUBAC can make K63 ubiquitin chains? The authors should also perform knockdown experiments to clearly rule out the role of cIAPs in Traf6-deficient cells*.

We did not propose that TRAF2/5 and LUBAC generate K63 ubiquitin chains in the absence of TRAF6. In fact, it is quite likely that linear and other types of ubiquitin chains generated in the absence of TRAF6 and Ubc13 could activate IKK and TBK1. This could explain how different TRAF proteins and LUBAC could compensate for each other in MAVS signaling in cells. Our in vitro assay largely recapitulates the K63 polyubiquitination-dependent pathways, but not the linear polyubiquitination-dependent pathway. How LUBAC mediates MAVS signaling in the absence of TRAF6 in vivo requires further investigation.

We have investigated the role of cIAPs in MAVS signaling using a small molecule SMAC mimetic compound (SM), which is known to bind cIAP1 and cIAP2 and induce their degradation. Treatment of WT, *Traf*6 KO and *Traf*2/*Traf*5 DKO cells with SM did not lead to any significant inhibition of IRF3 dimerization or induction of IFN? or IL6 by VSV (new Figure 2–figure supplement 1I–1L). Thus, cIAPs are largely dispensable for MAVS signaling under all conditions that we have tested. These results have been incorporated into the revised manuscript.

*3) The present study disagrees with several prior studies which suggested an essential role of TRAF3 in MAVS signaling. This is rather puzzling as the evidence presented previously was obtained independently by several groups. The authors should better investigate this discrepancy. Is this due to clonal variation and adaption among the TRAF KO cells? Are all different TRAFs expressed in KO cells as expected? The authors should also properly refer to those prior studies so that the contradiction can be more evident. For example, the paper does not give accurate reference to Oganesyan et al. (Nature 439:208). That paper is only referred as that TRAF3 is required for TLR-mediated IFN induction, though it did show that VSV activation of IRF3-dependent genes is markedly reduced in Traf3 KO MEFs*.

Our previous study using *Traf*3 knockout MEF cells showed that IFN? induction and IRF3 activation by Sendai virus were largely normal, except that at the later time point (9 hours after infection) there was a modest reduction of IFN? in the absence of TRAF3 ([54], Mol Cell 36: 315). In that study, we verified that TRAF3 was indeed not detectable in the *Traf*3 KO MEFs that we used. Using a different approach, here we showed that knockdown of TRAF3 by RNAi in *Traf*2/*Traf*5 DKO cells did not significantly impair IFN? or IL6 induction by VSV (Figure 2–figure supplement 1E–1G). Similarly, knockdown of Traf6 in *Traf*3-deficient MEF cells also did not inhibit IRF3 activation by Sendai virus (Figure 2–figure supplement 1H). Thus, unlike TRAF6 that functions redundantly with TRAF2/5, TRAF3 is not redundant with TRAF2/5 or TRAF6. Overall, although TRAF3 may play some roles in the induction of interferons in certain cells, it is largely dispensable for IRF3 activation under different experimental conditions. We have incorporated these new data into the revised manuscript. We have also revised the Discussion to leave open the possibility that TRAF3 may regulate interferon induction through a mechanism independent of regulating IRF3 activation.

*4) Another major concern is the large quantitative differences among the experiments: The levels of IL6 induction attained in*
Figure 2
*are in the range of 1000-fold, and in 2H of 60-fold. The levels of IFNβ induction in*
Figure 2
*approaches 100,000-fold while those in*
Figure 2*–figure supplement 1E are less than 600-fold. The author should clarify if these huge differences reflect just experimental variations or differences intrinsic to the cell clones used. The extent of the inter-clonal variation of response should be assessed by comparing a number of different cell clones of the same genotype. The authors should also clarify whether this large variation in extent of induction reflects differences in actual level of gene expression attained after viral infection or variation in the basal expression levels of the induced gene. In the latter case, the actual values of expression, rather than 'fold induction' should be presented. The exact nature of the differences in the values of gene induction in the various cells used in the same experiment is also not clear. Do the figures present average and standard deviation? In that case, average of what: different cell clones or several replicates of the same clone? In how many independent experiments? Are the various transfected proteins expressed at the same level and how do they compare to the endogenous levels? Western analyses and comparison of the expression levels of the transfected proteins should be presented*.

We apologize for this confusion. The large differences in the RNA levels as measured by q-RT-PCR were largely due to the variations in the basal levels (minus virus or 0 hour). For example, in Figure 2, the basal level of IL6 RNA in DKO + shTRAF6 + TRAF6-WT cells was ∼36 fold higher than the basal level in DKO-GFP control cells. This higher basal level was likely due to overexpression of TRAF6 in the cells. We have now modified Figure 2 by indicating the values of the IL6 RNA levels relative to the basal level of IL6 in the GFP control cells. It is apparent from these values that the ‘super-induction’ of IL6 in shTRAF6+TRAF6-WT cell was due to higher levels of TRAF6 in these cells.

The reviewers correctly pointed out the large differences in the folds of induction of IFN? in Figure 2 and Figure 2–figure supplement 1E. Such a difference was due to the use of different PCR machines which we acquired during the course of these experiments. The q-RT-PCR experiment in Figure 2–figure supplement 1E was performed using BioRad iQ5, whereas the experiment in Figure 2 was done using ABI ViiA7. The latter instrument has much higher sensitivity and allowed detection of very low levels of RNA in the samples. Since the basal levels of IFN? RNA were very low in uninfected cells, their detection was quite variable using different instruments. Such variability in the basal levels contributes to the large differences in the folds of induction presented in these figures. Despite such variation in different sets of experiments performed at different times, it is clear that virus infection induced very large and consistent increases of IFN? expression in the same set of experiments, allowing us to draw solid conclusions for each experiment.

With the exception of the SILAC experiments, the results presented in this paper are representative of at least two independent experiments. The error bars represent standard deviations of triplicate experiments. We have now indicated this fact in the Figure 2 legend with the following sentence:

“Unless indicated otherwise, error bars in this and other figures of this paper represent standard deviation of triplicate experiments.”